# Computation of the electroencephalogram (EEG) from network models of point neurons

**Pablo Martínez-Cañada**[1,2,3], **Torbjørn V. Ness**[4], **Gaute T. Einevoll**[4,5],
**Tommaso Fellin**[1,3], **Stefano Panzeri**[1,2]\*

**1** Neural Coding Laboratory, Istituto Italiano di Tecnologia, Genova, Italy, **2** Neural Computation Laboratory, Center for Neuroscience and Cognitive Systems @UniTn, Istituto Italiano di Tecnologia, Rovereto, Italy, **3** Optical Approaches to Brain Function Laboratory, Istituto Italiano di Tecnologia, Genova, Italy, **4** Faculty of Science and Technology, Norwegian University of Life Sciences, Ås, Norway, **5** Department of Physics, University of Oslo, Oslo, Norway

\* stefano.panzeri@iit.it

**Data Availability Statement:** The source code to simulate the LIF and multicompartment model networks can be downloaded at https://github.com/pablomc88/EEG_proxy_from_network_point_

## Abstract

The electroencephalogram (EEG) is a major tool for non-invasively studying brain function and dysfunction. Comparing experimentally recorded EEGs with neural network models is important to better interpret EEGs in terms of neural mechanisms. Most current neural network models use networks of simple point neurons. They capture important properties of cortical dynamics, and are numerically or analytically tractable. However, point neurons cannot generate an EEG, as EEG generation requires spatially separated transmembrane currents. Here, we explored how to compute an accurate approximation of a rodent's EEG with quantities defined in point-neuron network models. We constructed different approximations (or proxies) of the EEG signal that can be computed from networks of leaky integrate-and-fire (LIF) point neurons, such as firing rates, membrane potentials, and combinations of synaptic currents. We then evaluated how well each proxy reconstructed a ground-truth EEG obtained when the synaptic currents of the LIF model network were fed into a three-dimensional network model of multicompartmental neurons with realistic morphologies. Proxies based on linear combinations of AMPA and GABA currents performed better than proxies based on firing rates or membrane potentials. A new class of proxies, based on an optimized linear combination of time-shifted AMPA and GABA currents, provided the most accurate estimate of the EEG over a wide range of network states. The new linear proxies explained 85–95% of the variance of the ground-truth EEG for a wide range of network configurations including different cell morphologies, distributions of presynaptic inputs, positions of the recording electrode, and spatial extensions of the network. Non-linear EEG proxies using a convolutional neural network (CNN) on synaptic currents increased proxy performance by a further 2–8%. Our proxies can be used to easily calculate a biologically realistic EEG signal directly from point-neuron simulations thus facilitating a quantitative comparison between computational models and experimental EEG recordings.

neurons. Datasets generated in this study, scripts to plot the figures and weights of the CNN are available from http://doi.org/10.5281/zenodo.4506494.

**Funding:** This work was supported by the European Union's Horizon 2020 research and innovation programme under the Marie Skłodowska-Curie (grant agreement No 893825 to P.M.C), the NIH Brain Initiative (grants U19NS107464 to S.P. and T.F., and NS108410 to S.P.), the Simons Foundation (SFARI Explorer 602849 to S.P.), the European Union Horizon 2020 Research and Innovation Programme under Grant Agreement No. 785907 and No. 945539 [Human Brain Project (HBP) SGA2 and SGA3 to G.T.E.], and the Norwegian Research Council (NFR) through NOTUR - NN4661K to G.T.E. The funders did not play any role in the study design, data collection and analysis, decision to publish, or preparation of the manuscript.

**Competing interests:** The authors have declared that no competing interests exist.

## Author summary

Networks of point neurons are widely used to model neural dynamics. Their output, however, cannot be directly compared to the electroencephalogram (EEG), which is one of the most used tools to non-invasively measure brain activity. To allow a direct integration between neural network theory and empirical EEG data, here we derived a new mathematical expression, termed EEG proxy, which estimates with high accuracy the EEG based simply on the variables available from simulations of point-neuron network models. To compare and validate these EEG proxies, we computed a realistic ground-truth EEG produced by a network of simulated neurons with realistic 3D morphologies that receive the same synaptic input of the simpler network of point neurons. The new obtained EEG proxies outperformed previous approaches and worked well under a wide range of network configurations with different cell morphologies, distribution of presynaptic inputs, position of the recording electrode and spatial extension of the network. The new proxies approximated well both EEG spectra and EEG evoked potentials. Our work provides important mathematical tools that allow a better interpretation of experimentally measured EEGs in terms of neural models of brain function.

## Introduction

Electroencephalography is a powerful and widely used technique for non-invasively measuring neural activity, with important applications both in scientific research and in the clinic [1]. Electroencephalography has played a key role in the study of how both neural oscillations and stimulus-evoked activity relate to sensation, perception, cognitive and motor functions [2–4]. The electroencephalogram (EEG), like its intracranial counterpart, the local field potential (LFP), originates from the aggregation of all the electric fields generated by transmembrane currents across the surfaces of all neurons sufficiently close to the electrode [5–8]. The physics of how electromagnetic fields are generated from transmembrane currents is well understood, and mathematically described by forward models [6]. Yet, how to interpret changes in EEG across experimental conditions or diagnostic categories in terms of underlying neural processes remains challenging [1].

One way to better understand the EEG in terms of neural circuit mechanisms and to link theoretical models of brain functions to empirical EEG recordings is to compare EEG data with quantitative predictions obtained from network models. Network models of recurrently connected leaky-integrate-and-fire (LIF) point neurons are a current major tool in modelling brain function [9–11]. These models reduce the morphology of neurons to a single point in space and describe the neuron dynamics by a tractable set of coupled differential equations. These models are sufficiently simple to be understood thoroughly, either with simulations that are relatively light to implement, or by analytical approaches [12,13]. Despite their simplicity, they generate a wide range of network states and dynamics that resemble those observed in cortical recordings. They have been employed to satisfactorily explain a broad spectrum of different cortical mechanisms and cortical functions, such as sensory information coding [14,15], working memory [16,17], attention [18], propagating waves [19,20], non-rhythmic waking states [21,22], or the emergence of up and down states [23]. It remains an open question how to compute realistically EEGs from such widely used network models of simple point neurons.

A major problem in achieving the above goal is that in such LIF point neurons all transmembrane currents collapse into a single point in space and the resulting extracellular potential is, therefore, zero [6]. Previous studies comparing the simulation output of networks of

simple model neurons without a spatial structure with measures of graded extracellular potentials such as EEGs or LFPs have used ad-hoc approaches to estimate the EEG from variables available from simulation of the network, including the average membrane potential [23–28], the average firing rate [29–31], the sum of all synaptic currents [13,32,33], or the sum of absolute values of synaptic currents [14,34]. However, the limitations and caveats of using such ad-hoc simplifications to compute the EEG have been rarely considered and tested. As a result, it is still unclear how best to compute EEGs directly from the output of point-like neuron network models [35,36].

In order to generate extracellular potentials, spatially extended neuron models, i.e., multi-compartment neuron models, are required [37,38]. Previous studies have numerically computed the compound extracellular potential as the linear superposition of all single-cell distance-weighted transmembrane currents within a network of multicompartment neurons [39–41]. This approach is however computationally cumbersome, and it does not allow an easily tractable and exhaustive analysis of the dynamics of such networks. One alternative could be using a hybrid scheme [30,35,42,43] that projects the spike times generated by the point-neuron network onto morphologically detailed three-dimensional (3D) neuron models, and then computing the electric field generated by the currents flowing through these 3D neuron models. This scheme provides a simplification by separating the study of the network dynamics (described by the point-neuron network model) from that of field generation (described by the multicompartment neuron model), but still requires running cumbersome multicompartment model simulations for each simulation of the LIF network.

In this article, we implemented a much simpler and lighter method to predict the EEG based simply on the variables available directly from simulation of a point-neuron network model (e.g., membrane potentials, spike times or synaptic currents of the neuron models). We constructed several different candidate approximations (termed proxies) of the EEG that can be computed from networks of LIF point neurons. We then evaluated how well each proxy reconstructed a ground-truth EEG obtained when the synaptic input currents of the LIF model network were injected into an analogous 3D network model of multicompartmental neurons with realistic cell morphologies. This approach was shown to perform remarkably well in predicting the LFP [42], based on a specific weighted sum of synaptic currents from the point-neuron network model, for a specific network state (i.e., asynchronous irregular) of the LIF model network. However, the previously obtained LFP proxy did not include a head model that approximated the different geometries and electrical conductivities of the head necessary for computing a realistic EEG signal recorded by scalp electrodes. In this study, to compute the EEG, we chose a simple head model suitable for rodents. These animal species are the ones most commonly used in laboratories to invasively record neural activity. Studying rodent's EEG is thus directly relevant to interpret many available neuroscientific data, and it facilitates comparison of simulation results with empirical data. Additionally, we performed a proof-of-concept simulation of the ground-truth EEG and proxies on a complex human head model.

We derived a new proxy for the EEG that was validated against detailed simulations of the multicompartment model network, investigating different cell morphologies, variations of distribution of presynaptic inputs and changes in position of the recording electrode and in the spatial extension of the network model. Unlike previous studies which focused on approximations valid in specific network states [42], we also validated our proxies across the repertoire of network states displayed by recurrent network models, namely the asynchronous irregular (AI), synchronous irregular (SI), and synchronous regular (SR) [12] states, with different patterns of oscillations and individual cell activity. We found that a new class of simple EEG proxies, based on a weighted sum of synaptic currents, outperformed previous approaches,

including those optimized for predicting LFPs [14,42]. The new EEG proxies closely captured both the temporal and spectral features of the EEG. We also provided a non-linear refinement using a convolutional neural network to estimate the EEG from synaptic currents, which yielded moderate improvements over the linear proxy at the expense of increasing complexity of the EEG estimation model.

## Results

### Computing the ground-truth EEG and EEG proxies

We investigated how to compute a simple but accurate approximation of the EEG ("EEG proxy" hereafter) that would be generated by the activity of a LIF point-neuron network if its neurons had a realistic spatial structure. We therefore first simulated a well-established model of a recurrent network of LIF point neurons. We then fed the spiking activity generated by the LIF point-neuron network into a realistic 3D multicompartmental model network of a cortical layer and computed the EEG generated by this activity. We finally studied how to approximate this EEG simply by using the variables directly available from the simulation of the point-neuron network model.

The LIF point-neuron network was constructed using a well-established two-populations (one excitatory and one inhibitory) model of a recurrent cortical circuit [12], illustrated in Fig 1A. The network was composed of 5000 neurons: 4000 were excitatory (i.e., their projections onto other neurons formed AMPA-like excitatory synapses) and 1000 inhibitory (i.e., their projections formed GABA-like synapses). The neurons were randomly connected with a connection probability between each pair of neurons of 0.2. This means that, on average, the number of incoming excitatory and inhibitory connections onto each neuron was 800 and 200, respectively. The network receives thalamic synaptic input that carries sensory information and stimulus-unrelated inputs representing slow ongoing fluctuations of cortical activity. This type of network can generate a repertoire of different network states that map well into empirical observations of cortical dynamics [12,44]. Fig 1B shows, as an example, the asynchronous irregular spiking activity generated by a subset of the excitatory and inhibitory populations in response to a low firing rate of the thalamic input. We have shown in previous work that this model captured well (even more than 90% of the variance of empirical data) the dynamics of primary visual cortex under naturalistic stimulation [14,34,45].

We then computed a "ground-truth" EEG (referred to simply as "EEG" in the paper), following the hybrid modelling scheme [30,35,42,43], and used this ground-truth EEG to compare the performance of the different proxies. To do so, we created a network of unconnected multicompartment neuron models with realistic morphologies and homogeneous spatial distribution within the circular section of a cylinder of radius $r$ = 0.5 mm (Fig 1C), which roughly approximates the spatial extension of a layer in a cortical column. We focused on computing the EEG generated by neurons with somas positioned in a single cortical layer, layer 2/3 (L2/3), so that somas of multicompartment neurons are aligned along the same Z-axis coordinate (150 μm below the reference point Z = 8.5 mm). We chose to position somas in L2/3 based on previous computational work suggesting that this layer gives a large contribution to extracellular potentials [30,35]. The reference point Z = 8.5 mm was chosen to approximate the radial distance between the center of a spherical rodent head model and the brain tissue [46]. In this specific set of simulations performed for optimizing the proxies, we used the reconstructed morphology of a broad-tuft layer-2/3 pyramidal cell from rat somatosensory cortex available in the Neocortical Microcircuitry (NMC) portal [47,48], referenced as *dend-C250500A-P3_axon-C260897C-P2-Clone_9* (see "Methods"). We chose this pyramidal-cell morphology because its open-field geometry is expected to generate large extracellular potentials. Inhibitory

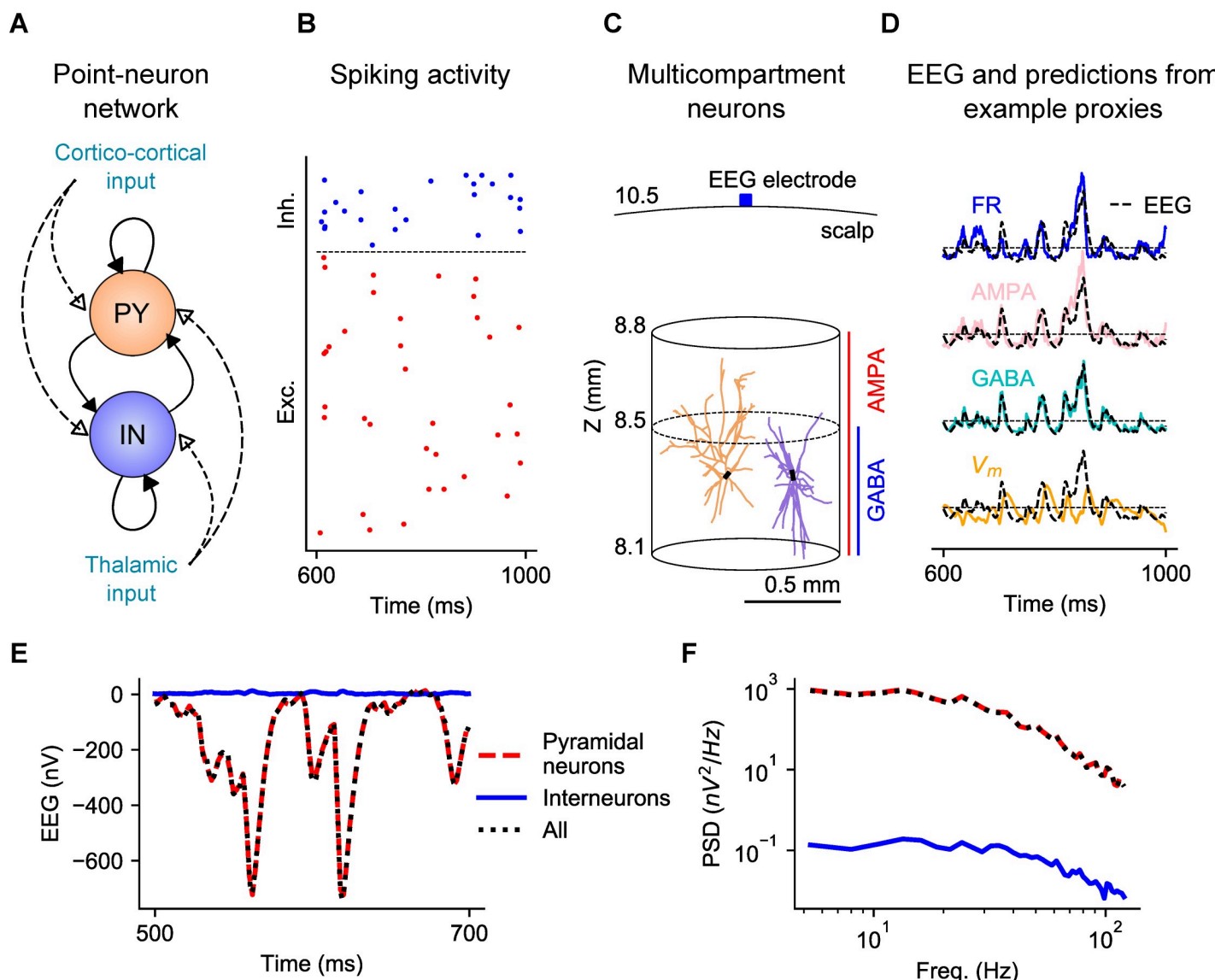

**Fig 1. Overview of the network models and computation of proxies and EEG.** (A) Sketch of the point-neuron network with recurrent connections between two types of populations: excitatory cells (pyramidal cells, PY) and inhibitory cells (interneurons, IN). Each population receives two kinds of external inputs: global ongoing cortico-cortical activity and thalamic stimulation. (B) Raster plot of spiking activity from a subset of cells in each population. (C) Sketch of the multicompartment neuron models used for generation of the EEG. Two representative model neurons are depicted, a pyramidal cell on the left and an interneuron on the right, positioned within a cylinder of $r = 0.5$ mm. While AMPA synapses are homogenously distributed over all compartments of both types of cells, GABA synapses on pyramidal cells are located only below $Z = 8.5$ mm. The EEG recording electrode is situated on the surface of the scalp layer. (D) Comparison between example proxies calculated from the point-neuron network and the ground-truth EEG computed from the multicompartment neuron model network. (E) EEG generated in the multicompartment neuron network by all neurons (dotted black), only pyramidal neurons (dashed red) or only interneurons (solid blue). (F) Corresponding power spectra for the three sets depicted in (E).

cells of the model were implemented using the morphology of L2/3 large basket cell interneurons (the most numerous class in L2/3 [47]).

AMPA synapses were homogenously positioned along the Z-axis in both cell types, representing uniformly distributed excitatory input. In our default setting, we assumed that all inhibitory synapses are made by large basket cell interneurons of the model, which based on their morphology would be principally located below the reference point $Z = 8.5$ mm. Thus, all

dendrites of inhibitory cells receive GABA synapses while only those dendrites of excitatory cells below Z = 8.5 mm receive GABA synapses, representing perisomatic inhibition.

EEGs were then generated from transmembrane currents of multicompartment neurons in combination with a forward-modelling scheme based on volume conduction theory [6]. To approximate the different geometries and electrical conductivities of the head, we computed the EEG using the four-layered spherical head model described in [35,49]. In this model, the different layers represent the brain tissue, cerebrospinal fluid (CSF), skull, and scalp, with radii 9, 9.5, 10 and 10.5 mm respectively, which approximate the dimensions of a rodent head model [46]. The values of the chosen conductivities are the default values of 0.3, 1.5, 0.015 and 0.3 S/m. The simulated EEG electrode was placed on the scalp surface, at the top of the head model (Fig 1C).

The time series of spikes of individual point neurons were finally mapped to synapse activation times on corresponding postsynaptic multicompartment neurons. Each multicompartment neuron was randomly assigned to a unique neuron in the point-neuron network and receives the same input spikes of the equivalent point neuron. Since the multicompartment neurons were not connected to each other, they were not involved in the network dynamics and their only role was to transform the spiking activity of the point-neuron network into a realistic estimate of the EEG. The EEG computed from the multicompartment neuron model network was then used as benchmark ground-truth data against which we compared different candidate proxies (Fig 1D).

## Dynamic states of network activity of the point-neuron network model

The LIF point-neuron network model we chose is known to generate a number of qualitatively different activity states [12,44] with patterns of variability of spike activity and network oscillations observed in cortical data. Here we recapitulate the different network states we generated for the LIF point-neuron network and that were used to evaluate the different proxies. The states generated by the LIF neuron network can be mapped by systematically varying across simulations the thalamic input ($\nu_0$) and the relative strength of inhibitory synapses ($g$). We then used three different measures to describe the network dynamics: synchrony, irregularity, and mean firing rate [12,44].

In Fig 2A, we plot these three descriptors as a function of $g$ and $\nu_0$. We individuated 3 different regions of the parameter space, each corresponding to a qualitatively different network state, according to the criteria employed by Kumar and collaborators [44]. The asynchronous irregular (AI) state is characterized by a low value of network synchrony ($< 0.01$), an irregularity level close to the value of a Poisson generator ($> 0.8$) and a very low firing rate, below 2 spikes/s. The synchronous irregular (SI) state has a level of network synchrony higher than that of the AI state (between 0.01 and 0.1), but with highly irregular firing of individual neurons (irregularity above 0.8). In the SI, neurons spike at low rate ($< 5$ spikes/s). For the synchronous regular (SR) state, the network exhibits high synchronous activity ($> 0.1$), a more regular single-cell spiking (irregularity below 0.8) and high spiking rate ($> 60$ spikes/s). Spike raster plots of excitatory and inhibitory cell populations of representative samples selected for each network state are shown in Fig 2B.

## Optimization and validation of proxies across different network states

We investigated how best to compute the proxy that combines the variables available directly from the simulation of a LIF point-neuron network model for accurately predicting the EEG over a wide range of network activity states. We explored different proxies that have been commonly used in previous literature for estimating the extracellular signal from point-neuron

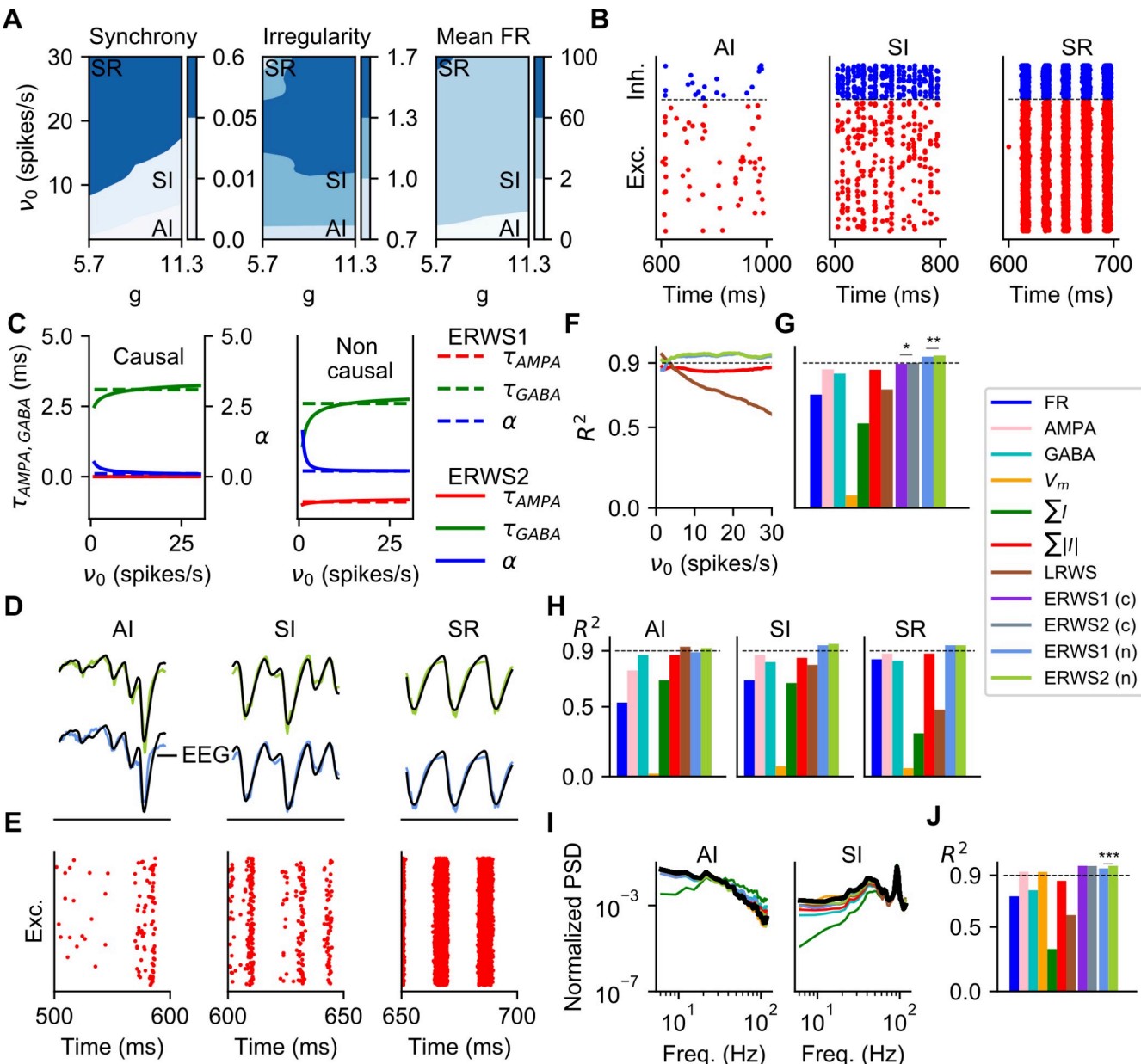

**Fig 2. Optimization and validation of proxies for different sets of network parameters ($v_0$, $g$).** (A) Dynamic states of network activity defined by the control parameters $g$ and $v_0$. The labels AI (asynchronous irregular), SI (synchronous irregular) and SR (synchronous regular) indicate the combinations of parameters that have been selected as representative samples of each network state. The synchrony and irregularity are unitless, the mean firing rate (FR) is measured in spikes/s. (B) Spiking activity from a subset of cells of the excitatory and inhibitory populations for the same samples shown in (A). (C) Optimized parameters of *ERWS1* and *ERWS2* (Eqs 7–9) as a function of the thalamic firing rate $v_0$. We considered two alternative scenarios. In the causal version of the proxy, the output depends only on present and past inputs so that the time delay parameters ($\tau_{AMPA}$ and $\tau_{GABA}$) are constrained to be positive. In contrast, non-causal proxies can be assigned positive or negative time delays. (D) Outputs of non-causal *ERWS1* (bottom row) and non-causal *ERWS2* (top row) proxies for different network states compared to ground-truth EEGs. (E) Spiking activity for the same simulation cases of panel D. (F) Average performance, evaluated by using the coefficient of determination $R^2$, of $\Sigma|I|$, *LRWS*, *ERWS1* (non-causal) and *ERWS2* (non-causal) calculated on the validation dataset as a function of $v_0$ (same colors as shown in (G)). The dotted line $R^2 = 0.9$ was chosen arbitrarily as a reference value of good performance and was used only for visual inspection of results. (G) Average $R^2$ of every proxy across all network instantiations $i$ of the validation dataset ($c$ is causal, $n$ is non-causal). The same colors shown in this legend are used throughout the article to identify the different proxies. Tests for statistical significance are computed only for the pair *ERWS1* (non-causal) and *ERWS2* (non-causal) and for the pair *ERWS1* (causal) and *ERWS2* (causal). (H) $R^2$ across network states. (I) Power spectral density (PSD) of the proxies and the EEG (in black). (J) Average $R^2$ computed across the 5–200 Hz frequency range of the $\log_{10}$(PSDs) of all network instantiations $i$ of the validation dataset.

networks: (i) the average firing rate (*FR*), (ii) the average membrane potential ($V_m$), (iii) the average sum of AMPA currents (*AMPA*), (iv) the average sum of GABA currents (*GABA*), (v) the average sum of synaptic currents ($\Sigma I$) and (vi) the average sum of their absolute values ($\Sigma |I|$). Furthermore, we propose here a new class of current-based proxies, (vii) the EEG reference weighted sum 1 (*ERWS1*) and (viii) the EEG reference weighted sum 2 (*ERWS2*), which are optimized linear combinations of time-delayed measures of AMPA and GABA currents. Indeed, an optimized weighted sum of synaptic currents (defined here as *LRWS*) was previously shown to be a robust proxy for the LFP [42]. The difference between *ERWS1* and *ERSW2* is that parameters of *ERWS2* adapt theirs values as a function of the strength of the external thalamic input $v_0$, whereas the parameters of *ERWS1* are not dependent on $v_0$ (see "Methods").

We only considered the transmembrane currents of pyramidal cells to generate the EEG (in the multicompartment neuron network) because the contribution of transmembrane currents of interneurons to the EEG was shown to be negligible (Fig 1E and 1F), in line with findings of Refs. [35] for the EEG and [42] for the LFP. Accordingly, we computed proxies of the LIF neuron network only using excitatory neurons. It is important to bear in mind, though, that interneurons, play an indirect role in generating the EEG in our models, because GABAergic currents in pyramidal cells depend on synaptic input from interneurons.

The firing rate of inhibitory neurons might be expected to contribute as well to the *FR* proxy and, as a consequence, to the EEG, as observed in Ref. [30]. To keep consistency with definition of the other proxies, we decided to compute the *FR* proxy based only on firing rates of excitatory cells. We checked that using a proxy computed on firing rates of both excitatory and inhibitory cells gave an EEG reconstruction accuracy considerably poorer than accuracy of the proxies based on synaptic currents (from proxy *iii* to proxy *viii* above).

The first 6 proxies taken from previous literature are parameter-free. The two new ones, *ERWS1* and *ERWS2* have 3 and 9 free parameters, respectively, which need to be optimized (Eqs 7–9). Following previous work [42], these parameters are the factor $\alpha$ describing the relative ratio between the two currents and a specific delay for each type of current ($\tau_{AMPA}$, $\tau_{GABA}$). We computed the values of these parameters by a cross-validated optimization of the predicted EEG across the different network states seen for the LIF model network.

For optimization (i.e., parameter training) of the proxies, we generated a large set of numerical simulations (522 simulations) by systematically varying the values of $g$ and $v_0$ over a wide state range. In each simulation instantiation, we set a given value $g$ and $v_0$ and used different random initial conditions (e.g., recurrent connections of the point-neuron network or soma positions of multicompartment neurons). The best-fit values of *ERWS1* and *ERWS2* were calculated by minimizing the sum of square errors between the ground-truth EEG and the proxy for all network instantiations of the optimization dataset (see "Methods", Eq 11).

We then cross-validated performance of the proxies by first computing how well they approximated EEGs generated by networks with the same properties that were used during proxy training, and then evaluating the proxies in networks in which we changed some of their features (e.g., cell morphologies or network size). Our reasoning was that if a proxy trained on some specific network features approximated well the EEG simulated across different network configurations, then we could hypothesize that our EEG proxies captured important and general properties of the relationship between the EEG and neural activity and thus could be used under a wide range of conditions.

Fig 2C shows the best-fit parameters ($\alpha$, $\tau_{AMPA}$ and $\tau_{GABA}$) found by the optimization algorithm for the two alternative scenarios considered here: causal and non-causal proxies (see also Table 1). For causal proxies, the predicted EEG depended only on present and past values of AMPA and GABA currents. Thus, the time delay parameters $\tau_{AMPA}$ and $\tau_{GABA}$ (quantifying

**Table 1. Parameters of *ERWS1* and *ERWS2*.**

| Proxy | Optimized values |
|---|---|
| *ERWS1* (causal) | $\tau_{AMPA}$ = 0 ms, $\tau_{GABA}$ = 3.1 ms, $\alpha$ = 0.1 |
| *ERWS2* (causal) | $a_1$ = 0, $b_1$ = 0, $c_1$ = 0, $a_2$ = -1.5, $b_2$ = 0.2, $c_2$ = 4, $a_3$ = 0.5, $b_3$ = 0.5, $c_3$ = 0 |
| *ERWS1* (non-causal) | $\tau_{AMPA}$ = -0.9 ms, $\tau_{GABA}$ = 2.3 ms, $\alpha$ = 0.3 |
| *ERWS2* (non-causal) | $a_1$ = -0.6, $b_1$ = 0.1, $c_1$ = -0.4, $a_2$ = -1.9, $b_2$ = 0.6, $c_2$ = 3, $a_3$ = 1.4, $b_3$ = 1.7, $c_3$ = 0.2 |

the delay by which the synaptic current contributes to the EEG) were constrained during optimization to be non-negative. For non-causal proxies, time delay parameters can take positive and negative values. Non-causal relationships between measured extracellular potentials and neural activity at multiple sites may emerge because of closed-loop recurrent interactions within the network [6]. The mathematical expressions of the optimized causal proxies ($\Phi_{ERWS1}(t)$ and $\Phi_{ERWS2}(t, v_0)$) are:

$$\Phi_{ERWS1}(t) = \sum\nolimits_{exc.} I_{AMPA}(t) - 0.1(\sum\nolimits_{exc.} I_{GABA}(t - 3.1 \text{ ms})), \tag{1}$$

$$\Phi_{ERWS2}(t, v_0) = \sum\nolimits_{exc.} I_{AMPA}(t) - (0.5 v_0^{-0.5})(\sum\nolimits_{exc.} I_{GABA}(t + 1.5 v_0^{-0.2} \text{ ms} - 4 \text{ ms})). \tag{2}$$

Expressions of the optimized non-causal proxies (where $v_0$ is unitless) are:

$$\Phi_{ERWS1}(t) = \sum\nolimits_{exc.} I_{AMPA}(t + 0.9 \text{ ms}) - 0.3(\sum\nolimits_{exc.} I_{GABA}(t - 2.3 \text{ ms})), \tag{3}$$

$$\Phi_{ERWS2}(t, v_0) = \sum\nolimits_{exc.} I_{AMPA}(t + 0.6 v_0^{-0.1} \text{ ms} + 0.4 \text{ ms}) - (1.4 v_0^{-1.7} + 0.2)(\sum\nolimits_{exc.} I_{GABA}(t + 1.9 v_0^{-0.6} \text{ ms} - 3 \text{ ms})). \tag{4}$$

For both *ERWS1* and *ERWS2*, in the non-causal versions, the time delay parameters were small (few milliseconds) but had opposite signs, $\tau_{GABA}$ was positive while $\tau_{AMPA}$ was negative (Fig 2C). In the causal version of both proxies, we observed a similar trend but $\tau_{AMPA}$ was constrained to 0 by the optimization. Thus, the best EEG proxies depend on past values of GABA synaptic currents and on current and future values of AMPA synaptic currents. These values are different from the optimal time delays ($\tau_{GABA}$ = 0 ms and $\tau_{AMPA}$ = 6 ms) found for the LFP in Ref. [42]. One reason for the observed difference between the previous LFP proxy and our new EEG proxies may relate to differences in spatial integration properties of the EEG signal and the LFP signal. Another probable cause of this difference is that in Ref. [42] the LFP proxy was optimized over a much smaller range of network states and external input rates ($v_0 < 6$ spikes/s). Indeed, our results for *ERWS2* show that optimal values of $\tau_{GABA}$ exhibit strong adaptation towards $\tau_{GABA}$ = 0 ms within the low regime of the external rate $v_0$. The parameter $\alpha$, which expresses the ratio of the contribution to the EEG of GABA relative to AMPA synaptic currents, also exhibits a strong adaptation. The dependence of $\alpha$ on the value of input rate $v_0$ in Fig 2C is particularly relevant because it reflects a larger weight of GABA currents for low values of $v_0$ and the opposite effect, stronger weight of AMPA currents, as the external rate increases.

We next explored the performance of proxies on networks with the same properties of those used for training (i.e., same network size and cell morphologies). To quantitatively evaluate the performance of all proxies, we computed for each proxy the coefficient of determination $R^2$, which represents the fraction of the EEG variance explained. The average $R^2$ calculated on the validation dataset (Fig 2G) shows a clear superiority of the new class of proxies. Both the causal and non-causal versions of *ERWS1* and *ERWS2* outperform all the other proxies, and the non-causal versions reach the best overall performance (*ERWS1*: $R^2$ = 0.94

and *ERWS2*: $R^2 = 0.95$). In agreement with previous results for the LFP [42], the three proxies that give the worst fits were *FR*, $\Sigma I$ and $V_m$.

To understand if the performance of proxies depended on the specific state of network activity, we first examined the performance of the most interesting proxies ($\Sigma|I|$, *LRWS*, *ERWS1* (non-causal) and *ERWS2* (non-causal)) separately for different values of the input rate $v_0$. We found that while *LRWS* performed well for low input rates (the range of external rates for which it was optimized [42]), its performance rapidly dropped with $v_0$ (Fig 2F). The other three proxies maintained a high $R^2$ for the whole spectrum of firing rates studied here, with *ERWS1* and *ERWS2* performing notably better than $\Sigma|I|$. Note also that *ERWS2* is the only proxy that yields a value of $R^2$ above 0.9 for all firing rates. We then computed the performance of these proxies separately for different types of network states. We found that the new proxies developed here, *ERWS1* and *ERWS2*, produced accurate fits of the EEG for all network states (Fig 2H), while accuracy of EEG approximations made by the other proxies was less uniform across network states.

The above analyses quantified how well the proxies approximated the actual values of the EEG in the time domain. We next examined how well the proxies approximated the overall power spectrum of the EEG rather than all variations of the EEG time series. In Fig 2I we show power spectral density (PSD) functions of all the proxies for the AI and SI states, compared to spectral responses of the EEG. In the whole frequency range considered (5–200 Hz), all proxies provided a qualitatively good fit of the EEG power spectrum, except $\Sigma I$, which attenuated low frequencies and amplified high frequencies. In Fig 2J we report the average $R^2$ computed for the $\log_{10}(\text{PSD})$ across all data points of the validation dataset. We logarithmically weighted the spectra to prevent $R^2$ to be dominated by low frequencies, neglecting the importance of errors at high frequencies. The performance obtained for power spectra confirmed the superiority of *ERWS1* and *ERWS2* also in the spectral domain. Interestingly, the average membrane potential ($V_m$), whose performance in the time domain was poor, performed instead better in the spectral domain. In contrast, the firing rate remained a poorly performing proxy also in the spectral domain.

## Time-shifted variants of proxies

The *ERSW* proxies were optimized for EEG prediction choosing optimal values for the time shifts between neural activity and the EEG. It is thus possible that the superior performance of the *ERWS* proxies over all others may have been due to the fact that the other proxies were not optimally time shifted. To investigate this hypothesis, we generated optimized time-shifted versions of all the other proxies by computing cross-correlation between the ground-truth EEG and all other proxies and choosing the optimum time shift of each proxy as the lag of the cross-correlation peak. We then compared the performance of the time-shifted versions of proxies in predicting the EEG with the performance of the *ERWS* proxies.

In this analysis, we recomputed the optimum time shift of every proxy separately for each network state, whereas the parameters of the *ERWS* proxies were jointly optimized (see previous section) over the entire simulated EEG dataset spanning all possible network states. Thus, this comparison was clearly favorable to the other proxies. Nevertheless, we still found that the *ERWS* proxies outperformed all previous proxies for the majority of network states. Only in the AI state, we observed that the *LRWS* proxy slightly outperformed *ERWS1* and *ERWS2*. The *ERWS2* proxy was the only one providing remarkably good performance across all states ($R^2 > 0.9$ over all states).

Further results came out of this analysis. Two proxies clearly improved the quality of their fits after time shifting, *FR* and $V_m$, but presented opposed time shifts: while *FR* was delayed,

$V_m$ was moved forward in time. A spike is a much faster event than the dipoles generated by synaptic activity and, as a result, a firing-rate proxy is expected to exhibit faster temporal changes than the EEG signal. By contrast, somatic integration of the postsynaptic membrane potentials following presynaptic spiking is a slower process that might lead to a signal more low-pass filtered than the EEG.

When comparing *AMPA* and *GABA* proxies, we observed that, in the AI state (Fig 3A), the temporal dynamics of the EEG signal was better approximated by the *GABA* proxy, whereas AMPA currents showed a faster response. Indeed, the performance of the *AMPA* proxy was improved after applying the corresponding time shift. As the firing rate of the external input increased and switched the network state from AI to SI (Fig 3B), the temporal evolution of the

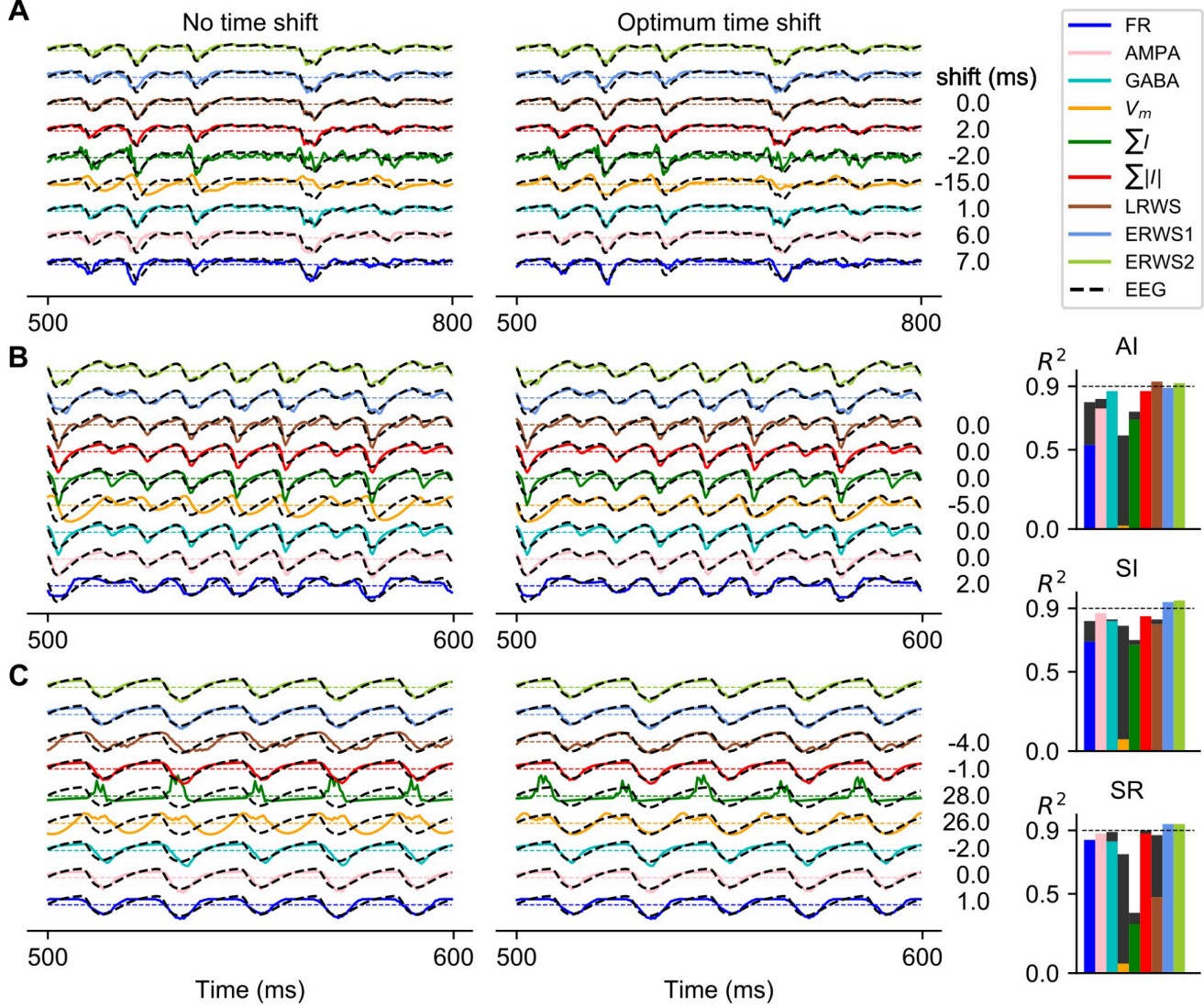

**Fig 3. Optimum time shift of proxies that maximizes cross-correlation with the EEG.** Comparison of the outputs of proxies and the ground-truth EEG before (left) and after (right) applying the optimum time shift, with the optimum time shift for each proxy and network state indicated on the right. Note that some proxies have positive time shifts for all network states (e.g., *FR*), while others (e.g., *GABA*) change the sign of the time shift when passing from the AI to the SR state. The network states shown are the following: AI in panel A, SI in panel B and SR in panel C. On the right: $R^2$ before (color bars) and after (black bars) applying the optimum time shift. *ERWS1* and *ERWS2* are not time shifted.

EEG began to diverge from GABA currents and, instead, AMPA currents were seen to better approximate the EEG.

## The performance of EEG proxies depends on the neuron morphology and distribution of synapses

Modelling studies have demonstrated that extracellular potentials generated by synaptic input currents vary with the neurons' dendritic morphology and the positions of individual synaptic inputs [6,50]. For example, morphological types that display a so-called *open-field* structure, such as pyramidal cells, have spatially separated current sources and current sinks that generate a sizable current dipole. Synaptic inputs onto neurons that have a *closed-field* configuration, such as interneurons, largely cancel out when they are superimposed so that the net contribution to the current dipole is weak [35]. The hybrid modelling scheme [30,35,42,43] gives us the opportunity to study, independently from the spiking dynamics of the point-neuron network, how different parameters of the multicompartment neuron network (e.g., distribution of synapses or dendritic morphology) affect the EEG signal and, as a consequence, modify the prediction capabilities of the proxies.

Above results (Figs 2 and 3) were computed using a specific multicompartmental model type of L2/3 pyramidal cell from rat somatosensory cortex (taken from the NMC database [47,48]) and referred as "NMC L2/3 PY, clone 9" (Fig 4A). Here, we studied whether the proxies derived for this morphology provided good approximations to the EEG generated by different cell morphologies. We thus quantified how well our proxies approximate the EEG generated by a different pyramidal-cell morphology taken also from rat somatosensory cortex ("NMC L2/3 PY, clone 0") and by a third morphology ("ABA L2/3 PY"), which is a L2/3 pyramidal cell from the mouse primary visual area [51]. It is important to note that the parameter values of proxies optimized for the morphology "NMC L2/3 PY, clone 9" were applied unchanged to the other morphologies across network states.

We found that *ERWS2* was the proxy with the highest prediction accuracy (Fig 4). It approximated extremely well the EEG across all three types of morphology and across all network states. The performance of both *ERWS* proxies in predicting the EEG generated by the mouse pyramidal neuron morphology ("ABA L2/3 PY", Fig 4, right column) was as good as the performance for the "NMC L2/3 PY, clone 9" morphology (probably because they have similar broad-tuft dendritic morphology, although different size). This suggests that the model generalizes reasonably well across species (at least for EEG generated by broad-tuft dendritic morphologies). *ERWS* proxies also performed well, though less compared to the morphology they were optimized for, on the EEGs generated by the other rat somatosensory cortex morphology ("NMC L2/3 PY, clone 0", Fig 4, middle column). The small decrease in performance was probably due to the fact that, unlike the broad dendritic tuft morphology used to optimized the proxy, this morphology incorporates long apical dendrites that separates AMPA synapses located in the tuft from GABA synapses more than 200 μm away. Analogously, the fact that the performance of *LRWS* did not decrease for the "NMC L2/3 PY, clone 0" morphology can be understood in terms of similarity between the pyramidal-cell morphology used to develop the *LRWS* proxy [42] and this morphology. The *LRWS* proxy [42] performed well across all morphologies in the AI state but its performance decreased across other states and morphologies. Other proxies performed poorly across different morphologies and/or states.

We also evaluated performance of proxies on the EEG generated by a heterogeneous population of pyramidal cells that had different morphologies (S1 Fig). In the same simulation, we randomly assigned the "NMC L2/3 PY, clone 9" morphology to half of the pyramidal-cell population and the "NMC L2/3 PY, clone 0" morphology to the other half. We found that

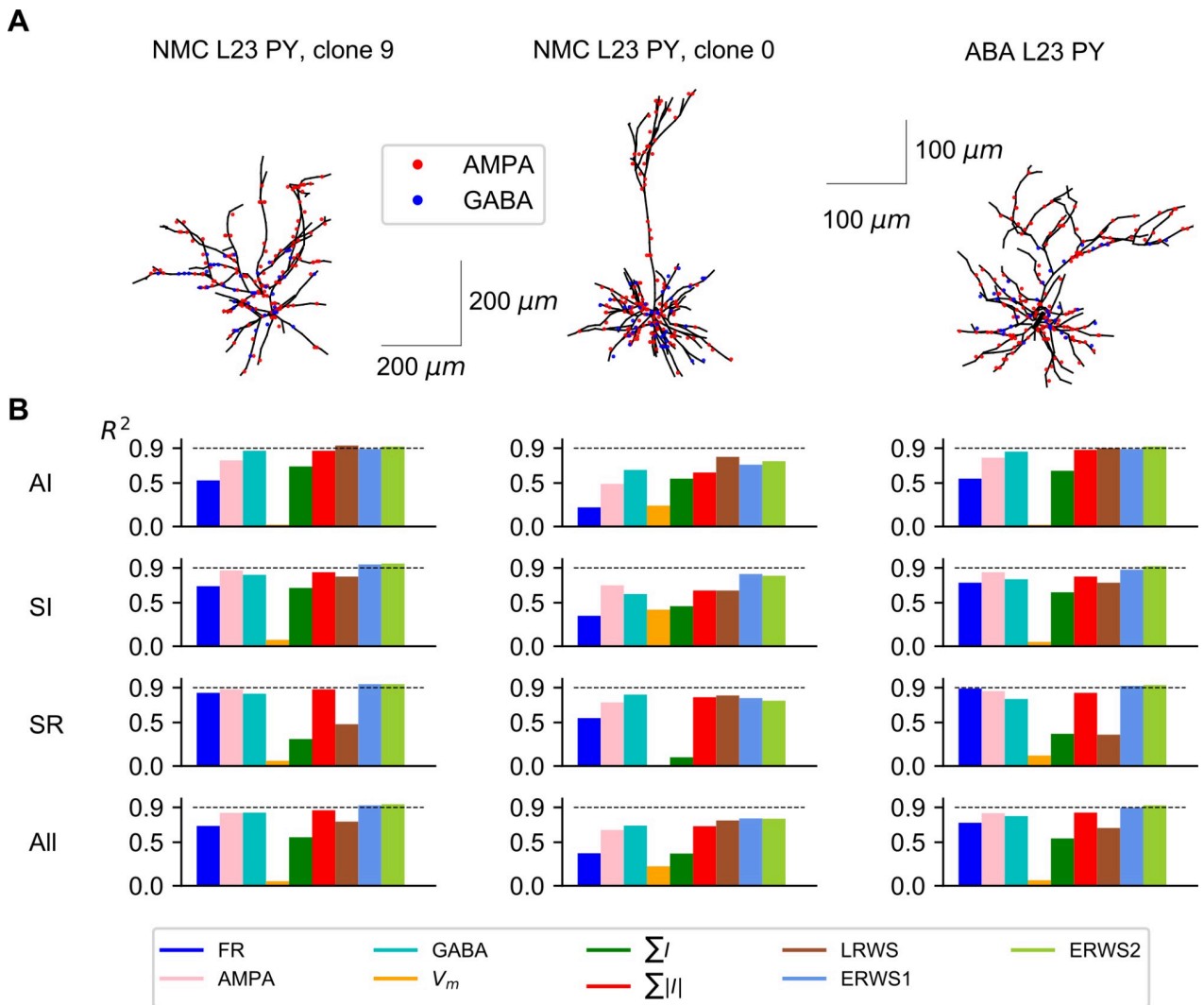

**Fig 4. Performance of proxies for different morphologies.** (A) Neuron reconstructions of L2/3 pyramidal cells acquired from the Neocortical Microcircuitry (NMC) portal [47,48] and the Allen Brain Atlas (ABA) [51]. For visualization purposes, in the synaptic distribution of each morphology, only a subset of AMPA and GABA synapses are shown, drawn randomly from all presynaptic connections. (B) $R^2$ computed for each morphology (columns) and network state (rows). The label "All" indicates the average $R^2$ across the three network states.

performance scores of proxies were between the values obtained independently for each morphology. This finding suggests that a mixed population of pyramidal cells, which includes a larger set of pyramidal-cell morphologies, could be well approximated by our proxies with an accuracy limited by the worst- and best-performing morphologies.

We next investigated how different spatial distributions of synapses on excitatory cells affect the performance of proxies (Fig 5). More specifically, GABA synapses were distributed on excitatory cells following two alternative approaches: located only on the lower part of the cell, primarily on the soma and basal dendrites ("Asymmetric") or homogeneously distributed across all dendrites ("Homogeneous"). Note that the "Asymmetric" case (Fig 5, left column) corresponds to default configuration shown in Fig 4A, left column ("NMC L2/3 PY, clone 9" morphology). The most significant change observed when distributing GABA synapses homogeneously on excitatory cells was an overall decrease of the performance of all proxies (but see

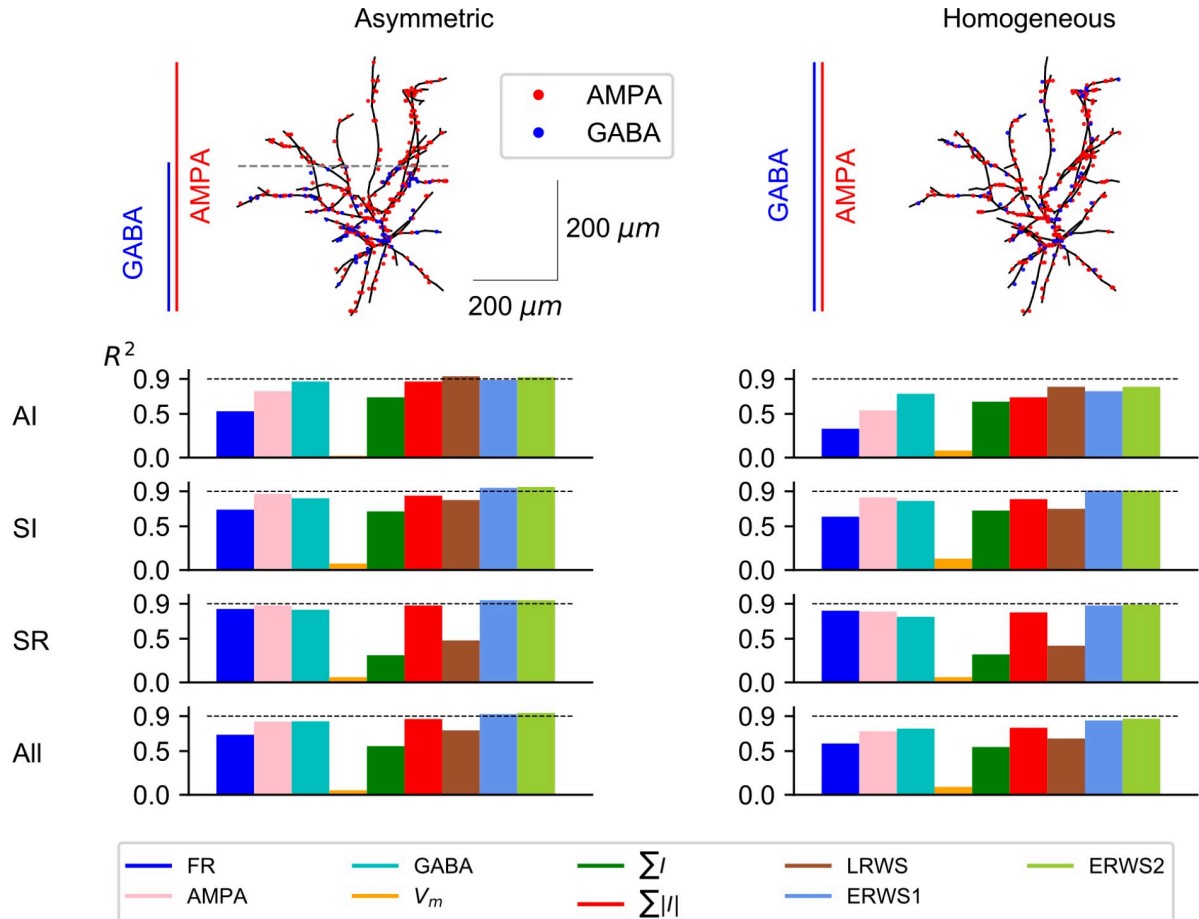

**Fig 5. Influence of synaptic distributions on performance of proxies.** Outline of the two different distributions of GABA synapses on excitatory cells: distributed only below the reference point $Z = 8.5$ mm ("Asymmetric") or distributed homogenously across all dendrites ("Homogeneous"). Each row below the diagram of model cells shows the corresponding $R^2$ for a different network state. The label "All" in the last row displays the average $R^2$ across the three network states.

$\Sigma I$), most prominently for the AI. These findings are in agreement with previous results obtained for the LFP proxy [42] in which a homogenous distribution of AMPA and GABA synapses on pyramidal cells resulted in the worst approximation of LFPs. In all scenarios, except for the AI state, *ERWS1* and *ERWS2* provided the best performance and their average $R^2$ values across network states reflect their superiority in both the asymmetric and homogenous distributions.

## Effects of the position of the electrode over the head model on the EEG and proxies

To investigate how the position of the electrode affects the EEG and performance of proxies, we simulated the EEG at four different locations over the head (Fig 6A). Simulation results, shown in Fig 6, are reported as a function of the angle between the electrode location and the Z-axis (angle Theta). We studied the effect of electrode position for the three different network states: AI, SI and SR. We first explored how properties of the EEG signal changed with the location of the electrode. As expected, the EEG amplitude, defined as the standard deviation of the EEG signal over time, decreased steeply when the electrode was moved away from the top

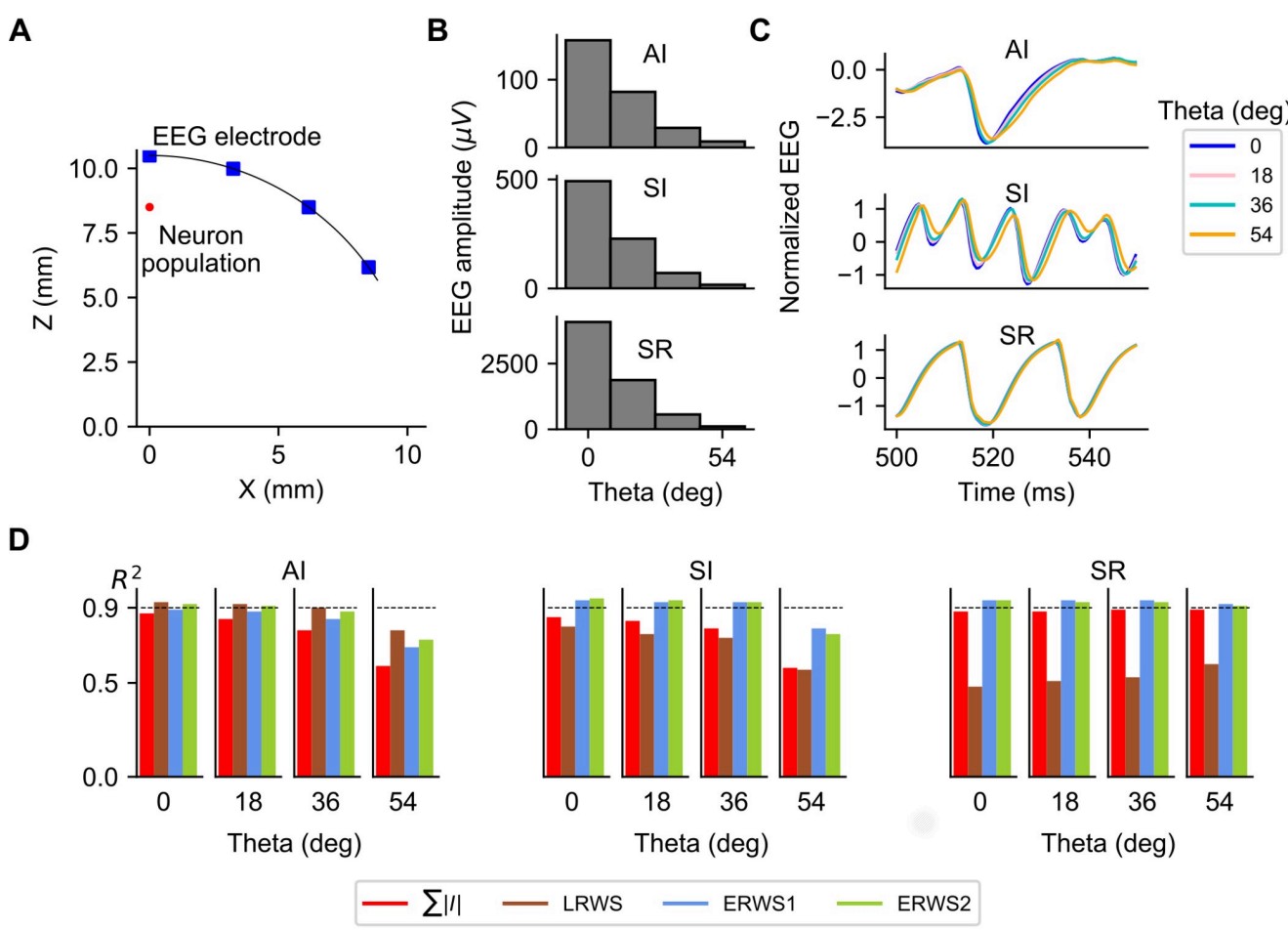

**Fig 6. EEG and proxies as a function of the position of the electrode over the head model.** (A) Illustration of the scalp layer in the four-sphere head model and locations where the EEG is computed. Location of the center of soma positions of the multicompartment neurons is marked as "Neuron population". (B) EEG amplitude, (C) normalized EEG and (D) performance of $\Sigma|I|$, *LRWS*, *ERWS1* and *ERWS2* as a function of the angle between the electrode location and the Z-axis (angle Theta), computed for the three different network states: AI, SI and SR.

of the head (Fig 6B). This decrease in EEG amplitude is consistent with previous simulation results of the 4-sphere head-model [35,39], in which a moderate attenuation of the EEG scalp potentials was observed when increasing the lateral distance from the center position along the head surface. Although the EEG amplitude is larger in the SR state, the relative variations of amplitude as a function of angle Theta were similar across network states. In contrast, we found (Fig 6C) sizeable differences in the normalized time courses of the EEG at different network states: an increase of angle Theta led to a delay of the EEG signal that was larger for the AI and SI states, but much weaker for the SR state. These results indicate that as the measurement point moves toward the zero-region of the current dipole, where the EEG power is much smaller, the signal-to-noise ratio is reduced and the influence of the high-frequency noise is more important. Since the signal power is significantly larger for the SR state, the effects of the high-frequency noise are less evident for the SR state.

Variations of properties of the EEG signal when the electrode was shifted from the top of the head affected the performance of proxies. As depicted in Fig 6D, in the AI and SI states the performance of $\Sigma|I|$, *LRWS*, *ERWS1* and *ERWS2* decreased when the angle Theta increased.

However, in the SR state the performance of proxies was largely independent of the position of the electrode, or it even increased with angle Theta in the case of the *LRWS* proxy. In any case, *ERWS1* and *ERWS2* gave the best performance in most scenarios, particularly *ERWS2* whose $R^2$ value was above 0.9, provided that Theta was smaller than 36 degrees.

## EEG estimation with a Convolutional Neural Network (CNN) proxy

The proxies considered above are all simple linear functions of the neural parameters of the LIF point-neuron network model. Linear proxies have the advantage of simplicity and interpretability. However, an alternative strategy for constructing an EEG proxy is training a convolutional neural network (CNN) to learn complex and possibly non-linear relationships between variables of the LIF point-neuron network model, such as AMPA and GABA currents, and the EEG. This could potentially improve the estimation of linear proxies, at the expense of increasing computational complexity and obscuring interpretation. Instead of using a deep neural network with many hidden layers that could largely increase complexity and prevent us from making any type of analogy with results of linear proxies, we opted for a simpler, shallow CNN architecture, with just one convolutional layer (Fig 7A). This CNN architecture was found to be sufficiently robust achieving a $R^2$ value of 0.99 on the test dataset (see Table 2). The network consists of one 1D convolutional layer ('Conv1D') with 50 filters and a kernel of size 20, followed by a max pooling layer ('MaxPooling1D') of pool size 2, a flatten layer and two fully connected layers of 200 units each one (marked as 'Dense' and 'Output' respectively). The input of the CNN is constructed by stacking data chunks of 100 ms (0.5 ms time resolution) extracted from the time series of AMPA and GABA currents, giving a 2 x 200 input layer.

The network was trained and tested on the same datasets (one for the training of the proxies, the other for the validation/testing of their accuracy) generated for optimization of parameters of the *ERWS1* and *ERWS2* proxies, using a first-order gradient descent method (Adam optimizer [52]) over 100 epochs (see "Methods"). In Fig 7B, we observe a quick convergence of the three metrics used to monitor training ($R^2$, MAE and MSE) towards optimal values ($R^2 \approx 1$, MAE $< 0.1$ and MSE $< 0.01$). Accuracy of predictions of the trained network, calculated on the test dataset, are shown in Fig 7. We computed the prediction error as the difference between amplitude values of the CNN-predicted and the true EEG signals. The probability distribution of the prediction error (Fig 7C) and the scatter plot of true versus predicted values (Fig 7D) both show a very accurate estimation of the EEG values. In Fig 7E, we illustrate some examples of predictions of the EEG signal compared to the ground-truth EEG for different network states. The best match between predicted and true EEG traces is seen for the SI state, although estimation performance remains high across all states.

The performance of the CNN was evaluated, like for the other proxies, as the average value of $R^2$ computed over all samples of the test dataset. As shown in Table 2 line A, the CNN clearly outperformed all other proxies on the test dataset and reached a very high performance score ($R^2 = 0.99$). We next assessed the performance of the CNN for the different configurations of the multicompartment neuron network, i.e., cell morphologies, distribution of presynaptic inputs and position of the recording electrode (Table 2 lines B, C and D). Compared to the best performing linear proxy, *ERWS2*, the CNN provided an increase of performance between 2 and 8% in most scenarios.

To gain insight into how AMPA and GABA inputs interact with layers of the network, we inspected the weights learned by different filters of the convolutional layer, as illustrated in S2 Fig for some examples of representative filters, depicted both in the time domain (panel A) and frequency domain (panel B). We observed that the majority of filters perform a band-pass

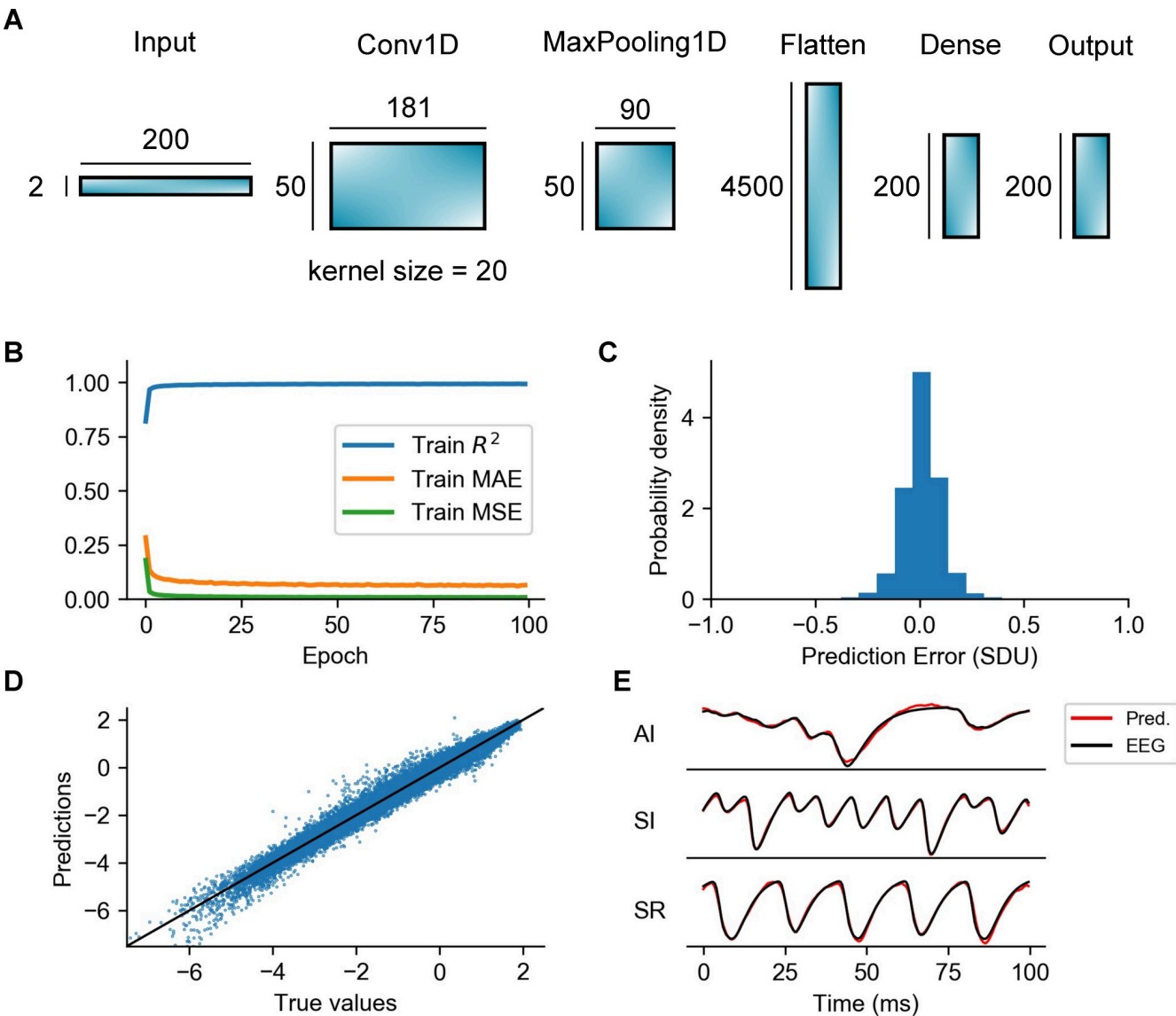

**Fig 7. Overview of the convolutional neural network, train errors and accuracy of EEG predictions.** (A) Illustration of the different types of layers included in the processing pipeline of the CNN architecture as well as the output shapes of each layer. Note that the 1D convolutional layer ('Conv1D') uses 50 filters and a 1D convolutional window (kernel) of size = 20. The total number of parameters of the entire CNN is 942450. (B) Training metrics collected during training: $R^2$, Mean Absolute Error (MAE) and Mean Squared Error (MSE). (C) Probability density function of the prediction error calculated on the test dataset. The error is expressed in Standard Deviation Units (SDU) (D) Predictions vs true values. Each dot of the scatter plot corresponds to amplitude values of the predicted and real EEG signals at a specific time step of the simulation. The continuous line represents a perfect EEG estimator. (E) Examples of predictions of the CNN compared to the ground-truth EEGs for different network states.

and high-pass filtering of AMPA and GABA inputs and their peak frequencies are within the range $[10^2, 10^3]$ Hz. This indicates that the CNN primarily uses the fast dynamics of the current inputs to construct an estimate of the EEG signal. We then asked whether we could disentangle the different transformation functions applied by the CNN to each type of input current. In signal processing, the impulse response of a linear system is typically used to understand the type of transfer function implemented by the system. The CNN included layers that were non-linear after the first convolutional layer. However, we could use a similar methodology to characterize the transformation function of the CNN by collecting the network

**Table 2. Performance (computed as $R^2$) of the CNN in comparison with $\Sigma|I|$, LRWS, ERWS1 and ERWS2 proxies.** The performance values shown for the test dataset (A) are averaged over all samples of the test dataset, while performance values in panels B, C and D are averaged over the samples of the different network states, i.e., AI, SI and SR.

| A: Performance on the test dataset | | | | |
|---|---|---|---|---|
| $\Sigma|I|$ | LRWS | ERWS1 | ERWS2 | CNN |
| 0.86 | 0.74 | 0.94 | 0.95 | 0.99 |

| B: Morphologies | | | | |
|---|---|---|---|---|
| Cell model | $\Sigma|I|$ | LRWS | ERWS1 | ERWS2 | CNN |
| NMC L2/3 PY, c. 9 | 0.87 | 0.74 | 0.92 | 0.94 | 0.97 |
| NMC L2/3 PY, c. 0 | 0.70 | 0.76 | 0.77 | 0.77 | 0.87 |
| ABA L2/3 PY | 0.85 | 0.67 | 0.90 | 0.92 | 0.94 |

| C: Distribution of synapses | | | | |
|---|---|---|---|---|
| Distribution type | $\Sigma|I|$ | LRWS | ERWS1 | ERWS2 | CNN |
| Asymmetric | 0.87 | 0.74 | 0.92 | 0.94 | 0.97 |
| Homogeneous | 0.77 | 0.65 | 0.83 | 0.87 | 0.89 |

| D: Position of the EEG electrode | | | | |
|---|---|---|---|---|
| Theta (rad) | $\Sigma|I|$ | LRWS | ERWS1 | ERWS2 | CNN |
| 0 | 0.87 | 0.74 | 0.92 | 0.94 | 0.97 |
| 0.31 | 0.86 | 0.74 | 0.91 | 0.93 | 0.97 |
| 0.63 | 0.82 | 0.72 | 0.90 | 0.91 | 0.96 |
| 0.94 | 0.69 | 0.68 | 0.80 | 0.81 | 0.87 |

responses to all possible combinations of unit impulses applied either to the AMPA or GABA inputs (S2C Fig). To extract a measure of the time shift applied by the network to AMPA and GABA inputs, we computed, for each unit impulse, the difference between the time when the impulse is applied and the time in which the absolute response of the network reaches its maximum. The histogram of time shifts applied to AMPA and GABA inputs (S2D Fig) shows that the CNN generally estimated the EEG signal by time shifting AMPA and GABA currents within the range [−2, 2] ms and the time shift could be either positive or negative.

## The new EEG proxies are robust to the addition of oscillatory power in canonical low-frequency bands

Magnetoencephalography (MEG), EEG and LFP studies consistently report that brain activity contains oscillations in canonical frequency bands that are superimposed upon an aperiodic broadband power-law spectrum [53–57]. An important question is how our proxies would perform in the presence of both oscillatory activity across different canonical EEG bands and of a broadband power-law spectrum.

Our spatially-unstructured local model of reciprocally connected excitatory and inhibitory populations could, when selecting appropriate parameters, intrinsically generate gamma rhythms (30–100 Hz) by the alternation of fast excitation and the delayed feedback inhibition [31,58]. As an example, Fig 2I reports EEG power spectra computed from the activity of our network model exhibiting gamma oscillations. Generation of gamma oscillations is an important model property because these oscillations are thought to underline important cognitive functions [59–61] and their disruption is associated to several cognitive disorders [62]. Additionally, we previously showed that our recurrent network model could also generate broadband power-law spectra that resembled cortical spectra [14, 42]. However, it could not intrinsically generate narrow-band oscillations in the canonical low-frequency bands, including delta (0.5–4 Hz), theta (4–8 Hz), alpha (8–12 Hz) and beta (15–30 Hz), all frequencies

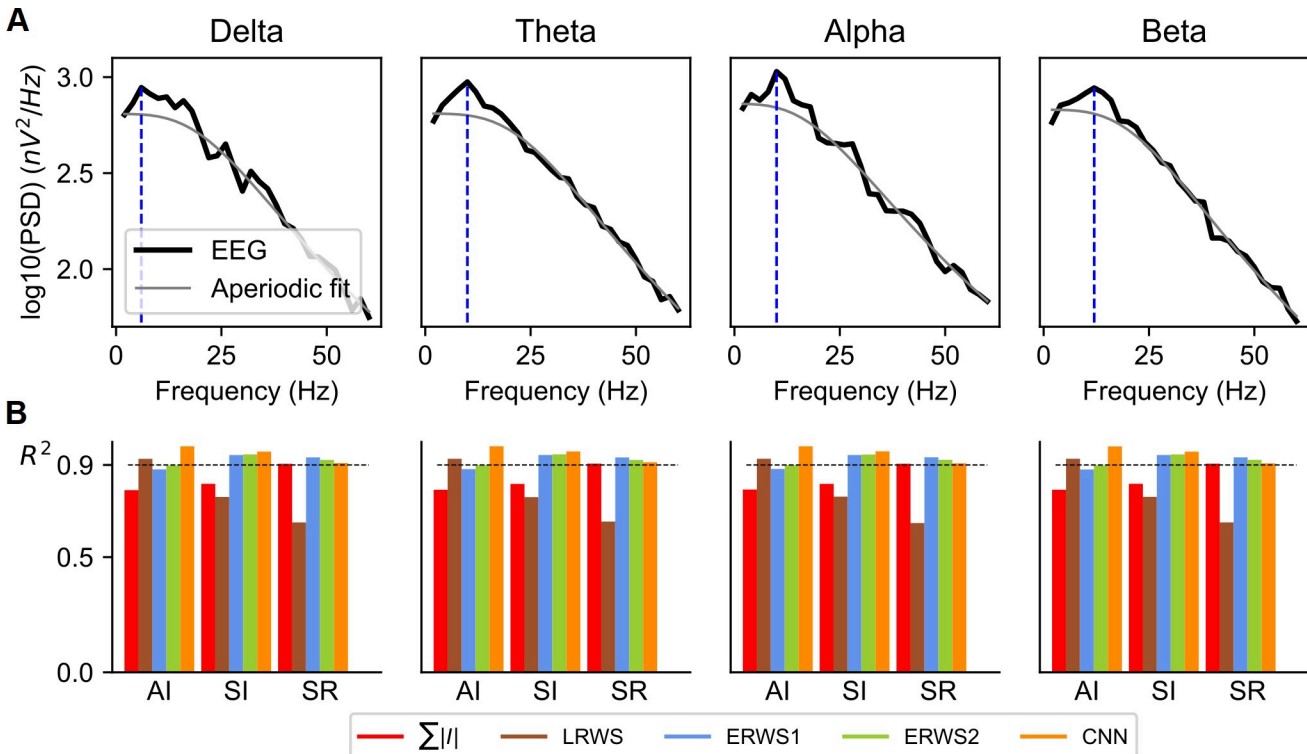

**Fig 8. Effect of adding oscillatory inputs in the canonical low-frequency bands.** (A) Power spectral density (PSD) functions of the ground-truth EEG for each one of the band-limited signals (delta, theta, alpha, and beta) added to the thalamic input of the model. The mean rate of the thalamic input was $v_0$ = 1.5 spikes/s. The fit to the simulated PSD of the aperiodic power-law component and of periodic components of the simulated EEG was performed using the FOOOF algorithm [53]. In the "aperiodic fit", the periodic band-limited component (obtained using a multi-Gaussian fit of the band-limited spectral peaks) was removed leaving only the aperiodic component. The blue dashed line indicates the position of the maximum value of the PSD used to compute the magnitude of the oscillatory peak. (B) Performance of $\sum|I|$, *LRWS*, *ERWS1*, *ERWS2* and CNN for the different band-limited inputs and network states (AI, SI and SR).

which are thought to arise from more complex loops (e.g. cortico-cortical and thalamocortical loops [63,64]).

In previous models, this type of oscillations in lower-frequency bands was generated by entrainment of an external lower-frequency input [14]. Therefore, to study effects of low-frequency oscillations on performance of EEG proxies, here we added band-limited components to the thalamic input, in either the delta, theta, alpha, or beta bands. As expected by our previous studies [14], the use of band-limited signals increased the oscillatory power of network activity in each of these canonical low-frequency bands (Fig 8A).

The amount of power to be added to the input in each canonical band was a free parameter in these simulations. To obtain reasonably realistic scenarios, we decided to set the added power such that the relative increase of oscillatory power in each frequency band with respect to the power of the aperiodic signal was similar to that observed in real data. To separate the narrow band oscillations from the aperiodic power-law spectral component, we applied the FOOOF algorithm [53] to our simulated EEG spectra. Using this algorithm, we computed the aperiodic-adjusted power (or flattened spectrum)—i.e., the magnitude of the oscillatory peak above the aperiodic component [53,54]. The aperiodic-adjusted power was computed by subtracting the aperiodic fit from the power spectrum (see "Methods"). We set parameters of the added power such that in all simulations the aperiodic-adjusted power values of the simulated EEG across the different frequency bands were in the range 0.05 to 0.2 (we manually explored

different amplitude values of power to be added, until the peak was at the desired amplitude in the EEG power spectrum). These values were in agreement with ranges of aperiodic-adjusted power values observed in recent studies in older adults that separate aperiodic and periodic features of power spectra in resting-state EEG [53] and MEG [54].

We computed the performance of the best EEG proxies ($\Sigma|I|$, *LRWS*, *ERWS1*, *ERWS2*, and the CNN) across the different network states when adding power to the thalamic input in the canonical low-frequency bands (Fig 8B). Interestingly, the prediction performance of the EEG proxies remained very similar to those we reported above when simulating the case in which the thalamic input was not modulated by a low-frequency oscillatory input (Fig 2 and Table 2). Overall, we found that *ERWS1*, *ERWS2* and the CNN produced accurate fits of the EEG for all network states, while accuracy of EEG approximations made by the other proxies was good for only one or two network states, but not for all. This suggests that our new proxies can work well to describe EEGs that show narrow-band peaks from the delta to the gamma range.

## Upscaling of the network model and integration of EEG by proxies calculated on local subnetworks of cells separated across space

The new EEG proxies were trained on EEGs simulated by a small localized population (5000 neurons within a radius of 0.5 mm). We then investigated whether our proxies could approximate well the EEG of a larger network model, and how the proxies could integrate neural activity across spatial locations to generate an accurate EEG.

We simulated an upscaled version of the network model that contained 20000 neurons (four times more neurons than the model used for proxy training). To preserve the density of neurons, we increased the circular area where cells were placed to cover a circular section of radius $r = 1$ mm. We also opted for preserving the number of synapses that each neuron receives by decreasing fourfold the probability of connection. In this way, we could expect that network dynamics of the upscaled model would remain largely unchanged. We divided this larger network into four spatially distinct subnetworks, each made of 5000 neurons. We then studied the performance of the EEG proxies in approximating the EEG generated by the extended network for different cases of connectivity between the subnetworks.

We first considered the simplest scenario of recurrent connectivity, where cells were all-to-all connected both within and across subnetworks (Fig 9A). The ground-truth EEG was computed by integrating the EEG signal from all 20000 cells in the multicompartment model network, whereas proxies were computed in two different ways: either by using only the activity of a local subnetworks of cells (S1, S2, S3 or S4), or by summing proxies of all subnetworks (S1 +S2+S3+S4). In either case, we observed that proxies locally calculated on subnetworks of cells (e.g., *ERWS1* and *ERWS2* in Fig 9B) could approximate very well the ground-truth EEG, as intuitively expected in an all-to-all connectivity scenario. Indeed, performance scores of global proxies (Fig 9C) were essentially identical to those of the proxies computed in individual subnetworks (Fig 9D). Another key observation was that prediction performance of both global and local proxies repeated the same trend of the 5000-neuron network used for training the proxies, with *ERWS1*, *ERWS2* and the CNN giving accurate EEG approximations in all network states. These results indicated that upscaling the network model preserved prediction performance, and that in an all-to-all connectivity configuration, computation of proxies from local subnetworks of cells separated across space could be used to produce reliable estimations of the EEG.

We then simulated a second network configuration in which recurrent connections of cells were constrained to target only cells of the same subnetwork (Fig 9E). To produce fully uncorrelated cortical dynamics between subnetworks, the external inputs to each subnetwork were

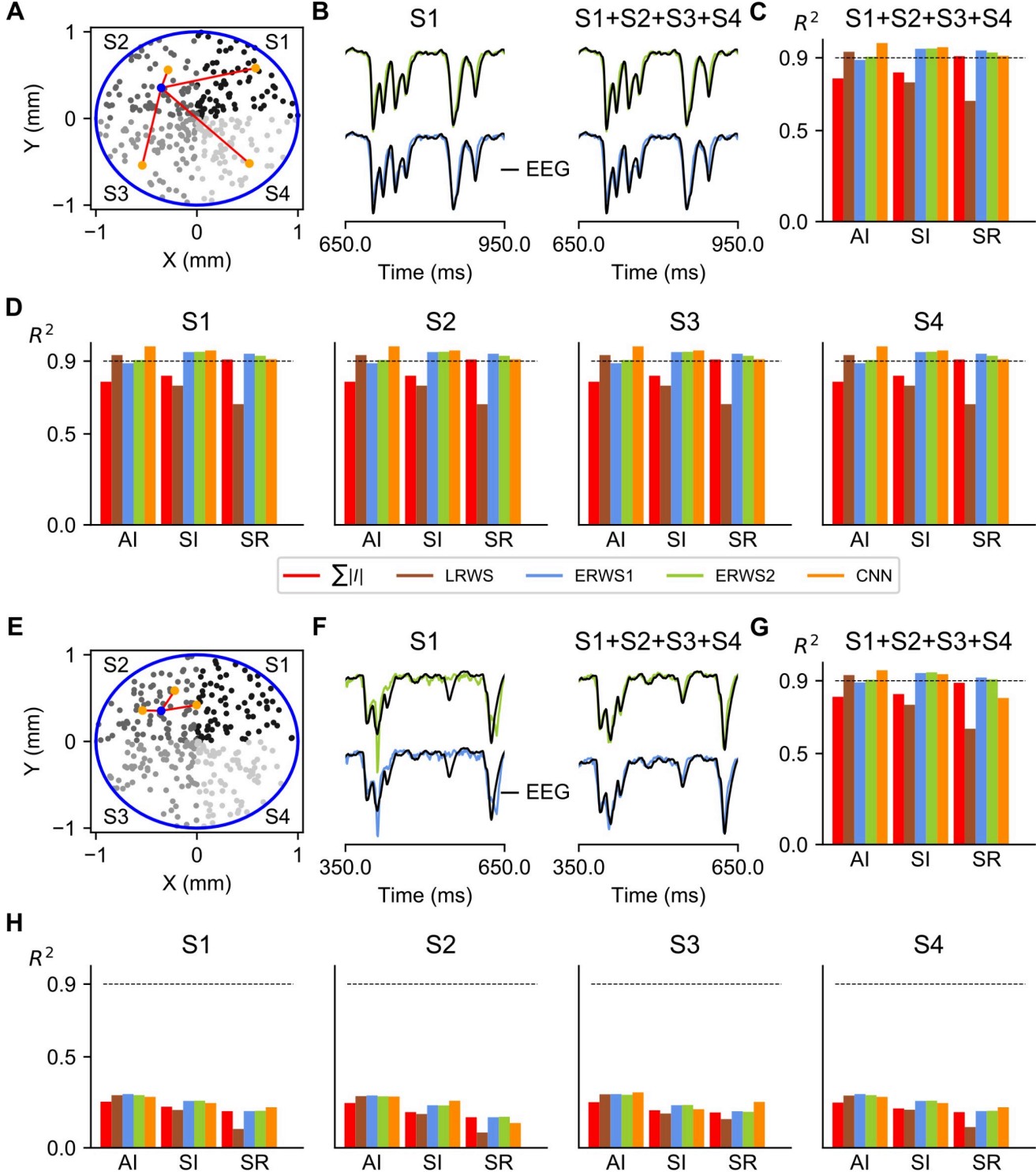

**Fig 9. Spatially extended network model.** (A) The network model was upscaled to cover a circular section of radius *r* = 1 mm. The number of cells of the original model was increased fourfold, although the upscaling preserved the densities of neurons and local synapses. The population of cells was divided in four subnetworks (S1, S2, S3 and S4) according to their position in each quadrant of the circular section. As an example, postsynaptic connections (red lines) of a randomly selected cell in S2 (blue spot) are depicted to illustrate the all-to-all connectivity pattern. Local proxies were computed for each subnetwork and the global proxy was calculated as the sum of proxies of all subnetworks (S1+S2+S3+S4). The ground-truth EEG was generated by summing contributions from all cells in the multicompartment model network. (B) Outputs of *ERWS1* (bottom row) and *ERWS2* (top row) proxies for the subnetwork S1 and of the global proxy compared with the ground-truth EEG. (C) Performance of global proxies. (D) Performance of proxies calculated

for the different subnetworks of cells. (E) In the second scenario, connectivity of each subnetwork is local as exemplified by postsynaptic connections of the selected cell in S2. (F) Outputs of *ERWS1* (bottom row) and *ERWS2* (top row) proxies for the subnetwork S1 and of the global proxy compared with the ground-truth EEG. Performance of global proxies (G) and proxies calculated for the different subnetworks of cells (H).

also independently generated. Unlike the model with all-to-all connectivity, we found that local proxies (S1, S2, S3 and S4) in the unconnected network could only partially approximate the EEG (Fig 9F and 9H), showing a significant decrease in performance in particular for faster network dynamics (i.e., SI and SR states). However, the linear sum of proxies computed on all subnetworks (S1+S2+S3+S4) approximated very well the EEG (Fig 9G), with a pattern of performance for all proxies similar to the one reported above for the smaller network (Fig 2) and to that observed in the upscaled model with all-to-all connectivity (Fig 9C).

Together, these results suggest that our new proxies can be successfully used to integrate neural signals across spatial locations to generate an accurate EEG in more spatially extended and larger network models.

## Prediction of the stimulus-evoked EEG

Evoked potentials are a useful technique that measures the transient response of the brain following presentation of a stimulus. Although the proxies we obtained have been optimized on long stretches of steady-state network activity, we investigated how well the proxies approximate an EEG evoked potential produced by a transient input. Fig 10 shows the spiking activity of the point-neuron network (panel A) and the ground-truth EEG (panel B) in response to a transient spike volley with a Gaussian rate profile applied to the thalamic input. This transient input simulates the thalamic input that reaches cortex when an external sensory stimulus is presented. A comparison of the performance obtained for all proxies is shown in panel C, while the outputs of *ERWS1*, *ERWS2* and the CNN are depicted in panel D, as an example, overlapped with the ground-truth EEG. We found that most of the current-based proxies

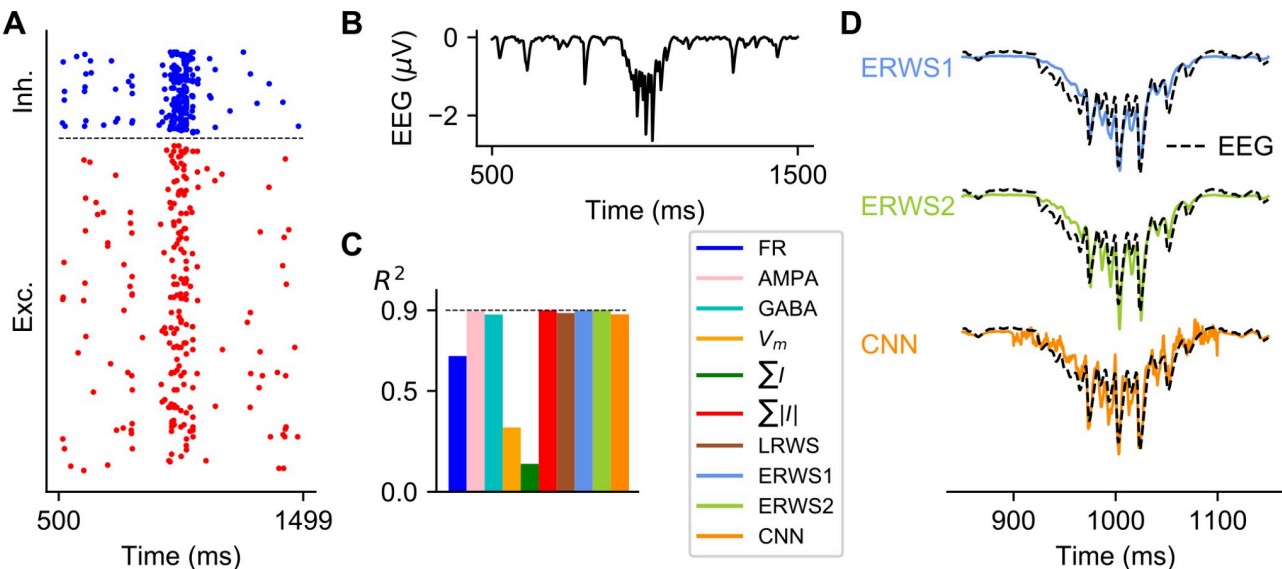

**Fig 10. Effect of transient activation of thalamic input with a Gaussian pulse packet.** (A) Raster plot of spiking activity from a subset of cells in each population in response to a transient spike volley with a Gaussian rate profile ($\sigma = 30$ ms) centered at 1000 ms. (B) Ground-truth EEG at the top of the head model. (C) Performance of proxies calculated between 850 and 1150 ms. (D) Outputs of *ERWS1*, *ERWS2* and the CNN compared to the ground-truth EEG.

approximated well the EEG when applying a transient burst of spikes of thalamic input, in particular $\Sigma|I|$, *ERWS1* and *ERWS2* which reached a performance of $R^2 = 0.9$. These results suggest that these types of proxies could also be employed to predict the type of transient response seen in evoked potentials.

### Evaluation of proxies in a human head model

Throughout this study, we have so far used a simple and analytically tractable head model (the four-sphere head model) to calculate EEGs. An important question is whether our proxies can work well when combined with more complex head models and, in particular, whether they can also be applied to the case of the human head. There are many high-resolution, anatomically detailed, and potentially personalized head models available, which for example take into account the folded cortical surface of the human brain [65–67]. As an initial exploration to evaluate applicability of our proxies to more realistic head models, in this Section we report a study of how our proxies approximate EEGs when computed using the New York head model [66], a very detailed model of the human head that incorporates 231 electrode locations (S4 Fig).

Head models for calculating EEG signals typically take current dipoles as input. When calculating the EEG signal with the four-sphere head model, we first calculated the current dipole moment of each individual cell, and then calculated all single-cell EEG contributions resulting from these current dipoles at their respective locations. However, it has been previously demonstrated [35] that for populations that are small compared to the distance to the EEG electrodes (~ 10–20 mm for human EEG recordings), the individual cell locations are relatively unimportant, and a negligible error is introduced by first summing all single-cell current dipoles into a population current dipole, and then calculating the EEG signal from this. We computed the population current dipole moments of two different subnetworks (S4D Fig) from the unconnected network model shown in Fig 9E. We then placed these current dipole moments in two different locations spatially separated and orthogonally oriented on the folded cortical surface of the New York head (S4B Fig). Topographic maps generated from EEG electrodes for the two dipoles are shown in S4C Fig, left column. The total EEG map was calculated as the sum of individual EEG maps generated by each current dipole.

We then investigated if our EEG proxies could be used to predict current dipole moments. We found that the EEG signal calculated at top of the head in the four-sphere head model, resulting from a current dipole directly below the electrode, was, in fact, just a scaling of the dominant component (the component aligned with the depth axis of the cortex) of the original current dipole (S3 Fig). This means that the proxies developed for the EEG signal at the top of the four-sphere head model, are in fact equally valid as proxies for the (normalized) dominant component of the population current dipole moment. Based on this observation, we generated EEG maps from our proxies (in our example, we used *ERWS2*) by plugging their outputs into the dominant component of the current dipole moment and recalculating EEGs on the New York head model. The EEG topographic maps resulting from the *ERWS2* proxy are depicted in S4C Fig, right column, which are shown to approximate qualitatively well the topographic maps of the ground-truth EEG data (S4C Fig, left column). The *ERWS2* proxy also predicted well time traces of the EEG signal at different electrode positions (S4 Fig, panels E and F).

Together, these results suggest that the EEG proxies developed here can be effectively and easily used in combination with complex models of the human head.

## Discussion

Interpreting experimental EEGs in terms of neural processes ultimately requires being able to compute realistic EEGs from simple and tractable neural network models, and then comparing

the predictions of such models with data. Here we contributed to the first goal by developing simple yet robust and accurate proxies to compute EEGs from recurrent networks of LIF point neurons, a model widely used to study cortical dynamics. The new proxies give very accurate reconstructions of both steady-state and transient EEGs over an extensive range of network states, different morphologies and synaptic distributions, varying positions of the EEG electrode and different spatial extensions of the network. These proxies thus provide a well validated and computationally efficient way for calculating a realistic EEG by using simple simulations of point-neuron networks.

## Simulations of EEGs using average membrane potentials or average firing rates are less accurate than those using synaptic currents

In many neural models, such as neural mass models [26,68], spiking network models [23,29,30] or dynamic causal models [25], EEGs or LFPs are simply modeled as the average firing rate or average membrane potential of excitatory neurons. While these assumptions are often reasonable, their effectiveness in describing the EEG has not been systematically validated. Here we found that these two established ways of computing the EEG worked reasonably well only under specific conditions. However, in agreement with previous results obtained for the LFP [14,42], we found that, for the EEG, proxies based on combinations of synaptic currents work much better and in more general conditions than proxies based on firing rates or membrane potentials. We found that, in several situations, membrane potential or firing rate proxies performed poorly. This was due to several factors. First, our work shows that the contribution of different proxies to the EEG is often time shifted and also state dependent (see below). Second, firing rates and membrane potentials have time scales different from those of transmembrane currents. Third, a proxy based only on the firing rates of excitatory neurons does not capture the contribution of inhibitory neurons, which is indirectly reflected in the EEG through the synaptic inputs of inhibitory neurons onto pyramidal cells.

Our results suggest that approximations of EEGs based on firing rates or membrane potentials of excitatory neurons should be discouraged, and replaced with the use of synaptic currents, whenever possible.

## State-dependent relationships between predicted EEG and neural activity

One important finding of our work was that the parameters of the best EEG proxy (*ERWS2*) were dependent on the cortical state of the network. This, in turn, implies that the relationship between the predicted EEG and the neural elements that originate it is state dependent. The state-dependent relationship of parameters of *ERWS2* may be due to properties of synaptic currents. We used realistic synaptic models, which were based on synaptic conductances and not on currents (as often implemented in simpler and less realistic models [12,31]). Unlike the current-based synapse model, AMPA and GABA currents of the conductance-based model depend on the membrane potential, which was shown in previous work to change with the external rate [69]. Consequently, the change of the membrane potential modifies the relative strength of AMPA and GABA currents and can modify, in turn, their contribution to the EEG. The optimization of parameters of *ERWS2* found a larger weight of GABA currents for low values of the external input and, conversely, a stronger weight of AMPA currents for higher values of the external input rate. This is in agreement with the increase of amplitude in AMPA postsynaptic potentials in comparison with GABA currents observed in previous studies [69].

Our results suggest that the contribution of neural activity to the EEG is a highly dynamic process, and highlight the importance of developing EEG proxies, such as those developed here to capture these variations.

## Robustness and generality of the EEG proxies across network states, cell morphologies, synaptic distributions and electrode locations

We found that, unlike all previously published proxies, our new optimized EEG proxies work remarkably well for a whole range of network states which capture many patterns of oscillations, synchronization, and firing regimes observed in neocortex [12]. Predicting well the EEG over a wide range of states is important because, in many cases, EEGs are experimentally used to monitor changes in brain states, and thus models used to interpret EEGs must be able to work well over multiple states. Our proxies were optimized using a specific pyramidal broad-dendritic-tuft morphology that generates large electric dipoles. We, however, showed that our proxies still had high accuracy when changing cell morphologies and distributions of presynaptic inputs. This suggests that our work, even though it could be improved using larger datasets of morphologies and synaptic distribution configurations, is already sufficiently general to capture the contribution to the EEG of some major types of pyramidal neurons. We also validated the performance of EEG proxies against changes in position of the recording electrode, with respect to the position chosen to train the proxies. The performance of proxies experienced only a moderate decrease as the position of the EEG electrode was shifted from the top of the head because of the progressive reduction in EEG amplitude. We finally demonstrated that our proxies, although trained on steady-state activity, can approximate well EEG evoked potentials, suggesting that our work could be relevant to model transient brain computations such as coding of individual stimuli or attentional modulations.

Previous work [42] used a similar approach based on optimizing a linear proxy to predict the LFP. We extended this work by computing the EEG, rather than the LFP. We used a head model that approximates the different geometries and electrical conductivities of the head, which was not necessary for the LFP proxy. Unlike this previous work, which considered only a reduced regime of network dynamics within the asynchronous or weakly synchronous states, we generated proxies trained and validated on a wider range of network states. Our EEG proxies were also validated on different pyramidal-cell morphologies reconstructed from experimental recordings, whereas the LFP proxy was validated on synthetically generated morphologies. As a result, our new optimized EEG proxies predict well the EEG over a wide range of states and different morphologies, unlike the LFP proxy, which was found in our study to work well only for a low-input-rate state and a specific morphology of pyramidal cells.

In sum, our new optimized EEG proxies provide a simple way to compute EEGs from point-neuron networks that is highly accurate across network states, variations of biophysical assumptions, and electrode position.

## Possible applications and impact of the new EEG proxies

Our work provides a key computational tool that enables applying tractable network models to EEG data with significant implications in two main directions.

First, when studying computational models of brain function, our work allows quantitative rather than qualitative comparison of how different models match EEG data, thereby leading to more objective validations of hypotheses about neural computations.

Second, our work represents a crucial step in enabling a reliable inference, from real EEG data, of how different neural circuit parameters affect brain functions and brain pathologies. Since the EEG conflates many circuit-level aggregate neural phenomena organized over a wide range of frequencies, it is difficult to infer from its measure the value of key neural parameters, such as for example the ratio between excitation and inhibition [1,70]. Our proxies could be used to develop tractable LIF neural networks that generate realistic EEG predictions from each set of neural network parameters. By fitting, in future work, such models to real EEG

data, estimates of neural network parameters (such as the ratio between excitation and inhibition or properties of network connectivity) could then be obtained from EEG spectra or evoked potentials. This approach could be used, for example, to test the role of excitation-inhibition balance in certain disorders [70–72], or to individuate neural correlates of diseases that show alterations of EEG activity [73–77]. Thus, our EEG proxies have clear promise for connecting EEG in human experiments to cellular and network data in health and disease. These future developments could complement other EEG modelling frameworks [68,78].

Although more work is needed to be able to interpret empirical EEGs in terms of network models, there are several facts that indicate that our proxies can be useful. Recent attempts to infer neural parameters from EEGs or other non-invasive signals, based on network models that use less accurate proxies than the ones developed here, are nevertheless beginning to provide credible estimates of key parameters of underlying neural circuit such as excitation-inhibition ratios [70,79], as well as accurate descriptions of cortical dynamics. For example, previous theoretical studies have modeled the LFP/EEG as the sum of absolute values of synaptic currents [14,15 34,45]. This type of proxy, though less accurate than those developed here, was sufficient to explain quantitatively several important properties of cortical field potentials, including the relationship between sensory stimuli and the spectral coding of LFPs [14], cross-frequency and spike-field relationships [34], and LFP phase of firing information content [15]. We thus expect that the new EEG proxies can build on these encouraging results and further improve the biological plausibility and robustness of neural parameter estimation from EEGs.

## Linear vs non-linear proxies

We developed both linear and non-linear EEG proxies based on synaptic currents. The linear proxies (*ERWS1* and *ERWS2*) were based on an optimized linear combination of time-shifted AMPA and GABA currents. Alternatively, we investigated the application of a shallow CNN that could capture more complex interactions between synaptic currents to estimate the EEG. Compared to the best performing linear proxy, *ERWS2*, the non-linear EEG proxy based on a convolutional network provided a sizeable increase of performance and it provided a very high performance in all conditions. The convolutional weights that we provide (Section "Data and Code Availability") can be used to easily compute these non-linear EEGs proxies using similar computational power as that employed for linear proxies. However, the drawback of CNNs is that it is harder to infer direct relationships between synaptic currents and the EEG, whereas these relationships are apparent and immediate to interpret with linear proxies. However, we showed that this problem could be in part attenuated when using tools to visualize the transformation function implemented by the CNN, which allow an understanding of how synaptic currents are transformed by the non-linear proxy.

## Limitations and future work

In our work, we made several simplifying assumptions regarding (i) the architecture and connectivity of the local recurrent network and of the thalamocortical and cortico-cortical loops, (ii) how to combine point-neuron and multicompartment networks, and (iii) the types of synaptic currents considered.

Our proxies have been extensively validated for a model with one class of pyramidal cells and are expected to be applied to models of any brain area in which the EEG is likely to be generated by one dominant population. We chose to model a single cortical layer, L2/3, based on previous computational work suggesting that this layer gives a large contribution to extracellular potentials [30,35]. Although we have shown that our proxies generalize well for different L2/3 pyramidal-cell morphologies, it will be important to extend our work to quantify

contributions from other cortical laminae and cell morphologies to the EEGs. Electrical potentials in the brain tissue add linearly and the superposition of individual contributions to the EEG is in principle straightforward to compute if the amplitude of each laminar contribution is known. Thus, it should be possible to approximate the total EEG by a suitable linear combination of individual proxies computed for each population. We envisage future studies that couple multi-layer spiking models of cortical circuits [30,80,81] with multi-layer multicompartment neuron models within the hybrid modelling scheme.

Our model of reciprocally connected excitatory and inhibitory populations can produce internally generated gamma oscillations. However, our simple recurrent network model cannot produce internally generated oscillations in the canonical lower-frequency bands (delta, theta, alpha, and beta) commonly observed in EEG and MEG. These slower oscillations likely arise from complex cortico-cortical and thalamocortical loops [63,64] which were not modeled in this study. We investigated the effect of such lower frequencies by superimposing low-frequency band-limited components to the synaptic input to the network. This simplified approach, however, cannot capture important aspects of cortico-cortical and thalamocortical loops, such as temporal synchronization and coupling between different brain areas, which could influence temporal properties of the EEG. Further studies are needed to test the validity of our proxies to approximate the EEG dynamics in realistic models of cortico-cortical and thalamocortical loops.

The hybrid modelling approach [30] offers the advantage that we can vary parameters of the EEG-generating model, e.g., cell morphologies or synaptic distribution, without affecting the spiking dynamics. The disadvantage of this approach is, however, that the multicompartment network does not match the point-neuron network in every respect. For instance, even though the synaptic input conductances were identical in the two models, the resulting soma potentials of multicompartmental neurons were not identical to those of the point neurons because of passive dendritic filtering or the lack of a membrane-voltage reset mechanism following spike, among other effects. This inconsistency could, at least partially, be resolved by extracting the effective synaptic weight distributions from multicompartment neurons and use them in the point-neuron network in order to make the two simulation environments even more similar [82].

Another limitation of our work is that it modelled only AMPA and GABA synapses and did not include NMDA synapses. An interesting topic for a future study would then be to extend the network models to include NMDA synapses and to analyze their impact on the current descriptions of the current-based proxies.

The connectivity of the recurrent network model used to train the proxies was random and distance-independent. The majority of local cortical connections are found within 500 μm [83]. The spatial scale of the decay of connection probabilities with distance is typically larger than a cortical column [80,81,84–86], which justified our simplified choice of distance-independent local network connectivity. When testing the proxies on spatially extended network models, we found our proxies to be accurate in two cases of distance-dependent connectivity (all-to-all connected or unconnected subnetworks). Further studies that use models with realistic distance-dependent connectivity are needed to provide accurate measures of the relative contribution of local proxies to the EEG as a function of cortical distance.

## Methods

### Overview of the approach for computing the proxies and the ground-truth EEG

Our focus is on computing an accurate prediction of the EEG (denoted as "proxy" in the following) based simply on the variables available directly from the simulation of a point-neuron

network model. The point-neuron network was constructed following a well-established configuration based on two populations of LIF point neurons, one excitatory and other inhibitory, with recurrent connections between populations [12], as illustrated in Fig 1A. The network receives two types of external inputs: a thalamic synaptic input that carries the sensory information and a stimulus-unrelated input representing slow ongoing fluctuations of cortical activity.

The ground-truth EEG (referred to simply as "EEG" in the paper) with which to compare the performance of the different proxies is here computed using the hybrid modelling scheme [30,35,42,43]. We created a network of unconnected multicompartment neuron models with realistic morphologies and distribute them within a cylinder of radius $r$ = 0.5 mm (Fig 1C). We focused on computing the EEG generated by neurons with somas positioned in one cortical layer so that the soma compartments of each cell are aligned in the Z-axis, 150 μm below the reference point $Z$ = 8.5 mm, and homogenously distributed within the circular section of the cylinder. In our default setting, all dendrites of inhibitory cells receive GABA synapses while only those dendrites of excitatory cells below $Z$ = 8.5 mm receive GABA synapses. AMPA synapses are homogenously positioned along the Z-axis in both cell types.

EEGs were generated from multicompartment neurons in combination with a forward-modelling scheme based on volume conduction theory [6]. From each multicompartment neuron simulation the current dipole moment of the cell was extracted with LFPy [39]. Next, these current dipole moments and the locations of the cells were used as input to the four-sphere head model to calculate all single-cell EEG contribution. The ground-truth EEG signal is the sum of all such single-cell EEG contributions. To approximate the different geometries and electrical conductivities of the head, we computed the EEG using the four-layered spherical head model described in [49]. In this model, the different layers represent the brain tissue, cerebrospinal fluid (CSF), skull, and scalp, with radii 9, 9.5, 10 and 10.5 mm respectively, which approximate the dimensions of a rodent head model [46]. The values of the conductivities chosen are the default values of 0.3, 1.5, 0.015 and 0.3 S/m. The EEG electrode is located on the scalp surface, at the top of the head model (Fig 1C).

The time series of spikes of individual point neurons were mapped to synapse activation times on corresponding postsynaptic multicompartment neurons. Each multicompartment neuron was randomly assigned to a unique neuron in the point-neuron network and received the same input spikes of the equivalent point neuron. Since the multicompartment neurons were not interconnected, they were not involved in the LIF network dynamics and their only role was to transform the spiking activity of the point-neuron network into a realistic estimate of the EEG. The EEG computed from the multicompartment neuron model network was then used as benchmark ground-truth data against which we compare different candidate proxies (Fig 1D).

## Definition and computation of the proxies that approximate the ground-truth EEG

A proxy (Φ) is defined as an estimation of the EEG based on the variables available from the point neuron model over all excitatory neurons. Unless otherwise stated, we only considered the contributions of pyramidal cells to generate the EEG (in both the point-neuron and multicompartment neuron networks). The first six proxies that we tested were those used in previous literature for predicting the EEG or the LFP from point-neuron networks. These were: the average firing rate (*FR*), the average membrane potential ($V_m$), the average sum of AMPA currents (*AMPA*), the average sum of GABA currents (*GABA*), the average sum of synaptic currents ($\Sigma I$) and average sum of their absolute values ($\Sigma|I|$). Note that $\Sigma I$ and $\Sigma|I|$ are defined as

the sum of both AMPA and GABA currents. Because of the opposite signs assigned to the AMPA and GABA currents, $\Sigma|I|$ is equivalent to the difference between these currents. Computation of the average *FR* was calculated with a temporal bin width of 1 ms, and then filtered with a 5-ms rectangular window to produce a smoother output of the *FR*.

For several reasons (e.g., different rise and decay time constants or different peak conductances), we expect that AMPA and GABA currents contribute differently to the EEG and that the optimal combination of both types of currents could involve different time delays between them. Following Mazzoni and colleagues [42], the new class of current-based proxies, the weighted sum of currents (*WS*), was based on a linear combination of AMPA and GABA currents, with a factor $\alpha$ describing the relative ratio between the two currents and a specific delay for each type of current ($\tau_{AMPA}$, $\tau_{GABA}$):

$$\Phi_{WS}(t) = \sum_{exc.} I_{AMPA}(t - \tau_{AMPA}) - \alpha\left(\sum_{exc.} I_{GABA}(t - \tau_{GABA})\right). \tag{5}$$

The optimal values of $\alpha$, $\tau_{AMPA}$ and $\tau_{GABA}$ were found to be 1.65, 6 ms and 0 ms for the LFP, respectively [42]. As a result, the LFP reference weighted sum (*LRWS*) proxy was defined as

$$\Phi_{LRWS}(t) = \sum_{exc.} I_{AMPA}(t - 6\text{ms}) - 1.65\left(\sum_{exc.} I_{GABA}(t)\right). \tag{6}$$

Here we also introduced two new proxies derived from the *WS* formulation: the EEG reference weighted sum 1 (*ERWS1*) and the EEG reference weighted sum 2 (*ERWS2*), whose parameters were optimized to fit the EEG under different network states of the point-neuron network. While the concept of *ERWS1* is similar to that of *LRWS*, with fixed optimal values of $\alpha$, $\tau_{AMPA}$ and $\tau_{GABA}$, the parameters of the *ERWS2* were defined as a power function of the firing rate of the thalamic input ($v_0$, unitless) to account for possible dependencies of the EEG with the external rate:

$$\Phi_{ERWS1}(t) = \sum_{exc.} I_{AMPA}(t - \tau_{AMPA(ERWS1)}) - \alpha_{ERWS1}\left(\sum_{exc.} I_{GABA}(t - \tau_{GABA(ERWS1)})\right), \tag{7}$$

$$\Phi_{ERWS2}(t, v_0) = \sum_{exc.} I_{AMPA}(t - \tau_{AMPA(ERWS2)}(v_0)) - \alpha_{ERWS2}(v_0)\left(\sum_{exc.} I_{GABA}(t - \tau_{GABA(ERWS2)}(v_0))\right), \tag{8}$$

$$\begin{aligned} \tau_{AMPA(ERWS2)}(v_0) &= a_1 v_0^{-b_1} + c_1, \\ \tau_{GABA(ERWS2)}(v_0) &= a_2 v_0^{-b_2} + c_2, \\ \alpha_{ERWS2}(v_0) &= a_3 v_0^{-b_3} + c_3. \end{aligned} \tag{9}$$

The total number of parameters to optimize was 3 for *ERWS1* ($\alpha_{ERWS1}$, $\tau_{AMPA(ERWS1)}$ and $\tau_{GABA(ERWS1)}$) and 9 for *ERWS2* ($a_1$, $b_1$, $c_1$, $a_2$, $b_2$, $c_2$, $a_3$, $b_3$ and $c_3$). We experimented with other classes of functions (e.g., exponential and polynomial functions) to describe the dependency of parameters of *ERWS2* with $v_0$ but the best performance results were found with a power function.

## Leaky integrate-and-fire point-neuron network

We implemented a recurrent network model of LIF point-neurons that was based on the Brunel model [31] and the modified versions developed in subsequent publications [14,15,34, 42,45,69]. These models have demonstrated to explain well and capture a large fraction of the variance of the dynamics of neural activity in primary visual cortex during naturalistic stimulation, including a wide range of cortical oscillations such as low-frequency (1–12 Hz) and

gamma (30–100 Hz) oscillations. In particular, the network structure and model parameters are the same ones used in [69] with conductance-based synapses (we refer the reader to this publication for an in-depth technical description of the implementation). Briefly, the network was composed of 5000 neurons, 4000 are excitatory (i.e., their projections onto other neurons form AMPA-like excitatory synapses) and 1000 inhibitory (i.e., their projections form GABA-like synapses). The neurons were randomly connected with a connection probability between each pair of neurons of 0.2. This means that, on average, the number of incoming excitatory and inhibitory connections onto each neuron was 800 and 200, respectively. Both populations received two different types of excitatory external input: a thalamic input intended to carry the information about the external stimuli and a stimulus-unrelated input representing slow ongoing fluctuations of activity. Spike trains of the external inputs are generated by independent Poisson processes. While the firing rate of every individual Poisson process for the thalamic input was kept constant in each simulation (within the range [1.5, 30] spikes/s), the firing rate of the cortico-cortical input was varied over time with slow dynamics, according to an Ornstein-Uhlenbeck (OU) process with zero mean:

$$\tau_n \frac{dn(t)}{dt} = -n(t) + \sigma_n \left( \sqrt{2\tau_n} \right) \eta(t) \tag{10}$$

Here $\sigma_n^2$ (0.16 spikes/s) is the variance of the noise, $\eta(t)$ is a Gaussian white noise and $\tau_n$ (16 ms), the time constant. The full network description is given in Tables 3 and 4, following the guidelines indicated in [87].

## Multicompartment-neuron network

The EEG was computed by projecting the spiking activity of the point-neuron network onto a network of multicompartment neuron models in which every multicompartment neuron is assigned a unique corresponding point neuron. A key factor for a successful representation of the EEG is selection of proper morphologies of multicompartment neurons with detailed and realistic dendritic compartments. Our focus was on computing the EEG for cortical layer 2/3 so that we acquired representative morphological reconstructions of L2/3 pyramidal cells and interneurons from publicly available repositories: the Neocortical Microcircuitry (NMC) portal [47,48] based predominantly on the data released by Markram and collaborators [47], and the Allen Brain Atlas (ABA) [51]. We also imposed our target animal model to be the rodent model. In our simulations, we evaluated three different types of morphologies of L2/3 pyramidal cells and one morphology of a specific type of L2/3 interneuron, the large basket cell interneuron (the most numerous class in L2/3 [47], represented as PY and LBC respectively in Table 5. Unless otherwise stated, the default morphology file used for pyramidal cells in our simulations is *dend-C250500A-P3_axon-C260897C-P2-Clone_9*.

Soma compartments of pyramidal cells and interneurons were randomly placed in a cylindrical section of radius 0.5 mm, at $Z$ = 8.35 mm. We assumed that GABA presynaptic inputs could only be located on dendritic compartments below the reference point $Z$ = 8.5 mm. AMPA synapses were homogenously distributed along the Z-axis in both cell types with random probability normalized to the membrane area of each segment. This configuration resulted in an asymmetric distribution of AMPA and GABA synapses onto pyramidal cells creating a stronger current dipole moment from these types of cells. Each multicompartment neuron was modeled as a non-spiking neuron with a passive membrane [38]. Tables 6 and 7 summarize properties of the multicompartment neuron network.

**Table 3. Description of the point-neuron network.**

| A: Model summary | |
|---|---|
| Structure | Excitatory-inhibitory (E-I) network |
| Populations | Two: excitatory and inhibitory |
| Input | 2 independent Poisson spike trains, one with a fixed rate and the other with a time-varying rate generated by an OU process |
| Measurement | Spikes, membrane potential, AMPA and GABA currents |
| Neuron model | Cortex: leaky integrate-and-fire (LIF) with fixed threshold and fixed absolute refractory time; external inputs: point process |
| Synapse model | Difference of exponential functions; conductance-based synapses |
| Topology | None |
| Connectivity | Random and sparse |

| B: Populations | | |
|---|---|---|
| Type | Elements | Size |
| Pyramidal cells | LIF neurons | 4000 |
| Interneurons | LIF neurons | 1000 |
| Thalamic input | Poisson generator | 1 |
| Cortico-cortical input | Poisson generator | 1 |

| C: Connectivity | | | |
|---|---|---|---|
| Name | Source | Target | Pattern |
| $AMPA_{Pyr\_Pyr}$ | Pyramidal | Pyramidal | Random convergent (p = 0.2), weight $g_{Pyr\_Pyr}$ |
| $AMPA_{Pyr\_Int}$ | Pyramidal | Interneuron | Random convergent (p = 0.2), weight $g_{Pyr\_Int}$ |
| $GABA_{Int\_Pyr}$ | Interneuron | Pyramidal | Random convergent (p = 0.2), weight $g_{Int\_Pyr}$ |
| $GABA_{Int\_Int}$ | Interneuron | Interneuron | Random convergent (p = 0.2), weight $g_{Int\_Int}$ |
| $AMPA_{tha\_Pyr}$ | Thalamic | Pyramidal | Fixed in-degree (800), weight $g_{tha\_Pyr}$ |
| $AMPA_{tha\_Int}$ | Thalamic | Interneuron | Fixed in-degree (800), weight $g_{tha\_Int}$ |
| $AMPA_{cort\_Pyr}$ | Cortical | Pyramidal | Fixed in-degree (800), weight $g_{cort\_Pyr}$ |
| $AMPA_{cort\_Int}$ | Cortical | Interneuron | Fixed in-degree (800), weight $g_{cort\_Int}$ |

| D: Neuron model | |
|---|---|
| Type | Leaky integrate-and-fire |
| Description | $\tau_m \frac{dV(t)}{dt} = -V(t) + V_{leak} - \frac{I_{tot}(t)}{g_{leak}}$, <br> $I_{tot}(t) = \sum_{N_{AMPA_{rec}}} I_{AMPA_{rec}}(t) + \sum_{N_{GABA_{rec}}} I_{GABA_{rec}}(t) + I_{AMPA_{ext}}(t)$, |

| E: Synapse model | |
|---|---|
| Type | Conductance-based synapse, difference of exponentials [31] |
| Description | $I_{syn}(t) = g_{syn} s_{syn}(t)(V(t) - E_{syn})$, <br> if a presynaptic spike occurs: <br> $s_{syn}(t) = \frac{\tau_m}{\tau_d - \tau_r}\left[ exp\left(\frac{-t-\tau_l}{\tau_d}\right) - exp\left(\frac{-t-\tau_l}{\tau_r}\right)\right]$ |

| F: Input | |
|---|---|
| Type | Description |
| Poisson generator | Thalamic input, time-constant input with rate $\nu_0$; each neuron receives 800 independent thalamic inputs |
| Poisson generator | Cortico-cortical input, OU process with zero mean; each neuron receives 800 independent cortico-cortical inputs |

| G: Global simulation parameters | |
|---|---|
| Simulation duration | 3000 or 10000 (only for Fig 8) ms |
| Temporal resolution | 0.05 or 0.1 (Figs 8 and 9) ms |
| Startup transient | 500 ms |

## Optimization and validation of EEG proxies

We created two different simulated datasets, one for optimization of the *ERWS1's* and *ERWS2's* parameters (Eqs 7–9), and the other dataset for validation of performance of all proxies. The datasets were generated by varying the two parameters of the point-neuron network

Table 4. Parameters of the neuron models used in the point-neuron network.

| A: Neuron model | | |
|---|---|---|
| **Parameter** | **Pyramidal cells** | **Interneurons** |
| $V_{leak}$ (mV) | -70 | -70 |
| $V_{threshold}$ (mV) | -52 | -52 |
| $V_{reset}$ (mV) | -59 | -59 |
| $\tau_{refractory}$ (ms) | 2 | 1 |
| $g_{leak}$ (nS) | 25 | 20 |
| $C_m$ (pF) | 500 | 200 |
| $\tau_m$ (ms) | 20 | 10 |
| **B: Connection parameters** | | |
| **Parameter** | **Pyramidal cells** | **Interneurons** |
| $E_{AMPA}$ (mV) | 0 | 0 |
| $E_{GABA}$ (mV) | -80 | -80 |
| $\tau_{r(AMPA)}$ (ms) | 0.4 | 0.2 |
| $\tau_{d(AMPA)}$ (ms) | 2 | 1 |
| $\tau_{r(GABA)}$ (ms) | 0.25 | 0.25 |
| $\tau_{d(GABA)}$ (ms) | 5 | 5 |
| $\tau_l$ (ms) | 1 | 1 |
| $g_{AMPA(rec.)}$ (nS) | 0.178 | 0.233 |
| $g_{AMPA(tha.)}$ (nS) | 0.234 | 0.317 |
| $g_{AMPA(cort.)}$ (nS) | 0.187 | 0.254 |
| $g_{GABA}$ (nS) | 2.01 | 2.7 |

commonly used for exploration of different network states [31,44]: the rate of the external input, $v_0$, and the relative strength of inhibitory synapses, defined here as $g = g_{Int\_Pyr}/g_{Pyr\_Pyr}$. We selected 58 values of $v_0$ within the range [1.5, 30] spikes/s and 3 values of $g$ (5.65, 8.5 and 11.3), which encompass the different network states: asynchronous irregular, synchronous irregular and synchronous regular [12]. For every pair $(v_0, g)$, we generated three simulations of the point-neuron and multicompartment-neuron networks with different random initial conditions (e.g., recurrent connections of the point-neuron network or soma positions of multicompartment neurons). The simulated outputs from two of these network instantiations were used for the optimization dataset and the other one for the validation dataset.

Prior to comparing the EEG traces with the point-neuron model predictions, we z-scored the proxies and the EEG signal by subtracting their mean value and dividing by the standard deviation. The best parameters of *ERWS1* and *ERWS2* were calculated by minimization of the sum of the square errors *SSE* between the ground-truth EEG and the proxy for all network instantiations $i$ of the optimization dataset:

$$SSE = \sum_i \sum_t (EEG_i(t) - \Phi_i(t))^2 \tag{11}$$

Table 5. Morphologies types and file identifiers used in the multicompartment neuron network model.

| Cell type | Animal species | File identifier | Source |
|---|---|---|---|
| L2/3 PY | Rat | dend-C250500A-P3_axon-C260897C-P2-Clone_9 | NMC |
| L2/3 PY | Rat | dend-C260897C-P3_axon-C220797A-P3-Clone_0 | NMC |
| L2/3 PY | Mouse | Cux2-CreERT2, ID:486262299 | ABA |
| L2/3 LBC | Rat | C250500A-I4_Clone_0 | NMC |

**Table 6. Description of the multicompartment neuron network.**

| A: Model summary | |
|---|---|
| **Structure** | Unconnected populations of multicompartment neurons |
| **Populations** | Two: pyramidal cells and interneurons |
| **Input** | Presynaptic spiking activity as modeled by the point-neuron network |
| **Measurement** | EEG, current dipole moment |
| **Neuron model** | Multicompartment neuron model based on the passive cable formalism |
| **Synapse model** | Difference of exponential functions; conductance-based synapses |
| **Topology** | Cylindrical volume with radius $r = 0.5$ mm |
| **Connectivity** | None |
| **B: Populations** | |
| **Type** | Populations of 4000 pyramidal cells and 1000 interneurons |
| **Cell positions** | Soma compartments located at $Z = 8.35$ mm and randomly distributed within the circular section of the cylinder |
| **Cell orientations** | Fixed orientation with apical dendrites oriented along the Z-axis |
| **Morphologies** | Reconstructed morphologies from the NMC and ABA (Table 5); axons removed if present |
| **C: Connectivity** | |
| No network connectivity, synaptic inputs are generated by the point-neuron network with the same synaptic parameters (Table 4) | |
| **D: Neuron model** | |
| **Type** | Multicompartment reconstructed morphologies |
| **Description** | Non-spiking neurons based on the passive cable formalism (except in subsection "The performance of EEG proxies depends on the neuron morphology, distribution of synapses and the type of dendritic conductances"), with membrane capacity $c_m$, membrane resistivity $r_m$, axial resistivity $r_a$ and leak reversal potential $E_L$. |
| **E: Synapse model** | |
| **Type** | Conductance-based synapse, difference of exponentials |
| **Description** | $I_{syn}(t) = g_{syn}s_{syn}(t)(V(t)-E_{syn})$, <br> $s_{syn}(t) = A\left[exp\left(\frac{-t-\tau_l}{\tau_d}\right) - exp\left(\frac{-t-\tau_l}{\tau_r}\right)\right]$, <br> where $A$ is a normalization factor to give a peak conductance $g_{syn}$ |
| **F: Input** | |
| **Type** | Spike times of spiking neuron network (including thalamic and cortico-cortical input spikes), no recurrent input |
| **Description** | All dendrites of interneurons receive GABA synapses while only those dendrites of pyramidal cells below $Z = 8.5$ mm receive GABA synapses; AMPA synapses are homogenously positioned along the Z-axis in both cell types; synapse locations are randomly assigned onto cell compartments assuming a probability proportional to the compartment's surface area divided by the total surface area of the cell |
| **G: Global simulation parameters** | |
| **Simulation duration** | 3000 or 10000 (only for Fig 8) ms |
| **Temporal resolution** | 0.05 or 0.1 (Figs 8 and 9) ms |
| **Startup transient** | 500 ms |

Time constants of proxies (Eqs 7–9) were restricted to be discrete variables as the simulation time is a discrete variable. This turns the optimization problem into a discrete optimization problem, which is harder to solve than a continuous optimization problem. However, the limited number of parameters that need to be optimized allowed us to run a simple brute-force parameter search.

The performance of each proxy was evaluated by using the coefficient of determination $R^2$, which is the fraction of the EEG variance explained by the proxy. $R^2$ is computed as the

**Table 7. Parameters of multicompartment neurons.**

| Parameter | Pyramidal cells | Interneurons |
|---|---|---|
| $c_m$ (μF/cm$^2$) | 1 | 1 |
| $r_m$ (kΩcm$^2$) | 30 | 20 |
| $r_a$ (Ωcm) | 100 | 100 |
| $E_L$ (mV) | -70 | -70 |

squared value of the correlation coefficient. The validation results were calculated based on the average $R^2$ of every proxy across all network instantiations $i$ of the validation dataset.

## Implementation of the convolutional neural network

The processing pipeline of the CNN architecture, illustrated in Fig 7A, was based on the machine-learning library Keras running on top of TensorFlow [88]. The CNN consists of a one-dimensional (1D) convolutional layer with 50 filters and a kernel of size 20, followed by a max pooling layer of pool size 2, a flatten layer and two fully connected layers of 200 units each (one of them is the output layer). The rectified linear unit (ReLU) function was used as the activation function for all layers, except for the output layer. To reduce overfitting, we applied L2 activity regularization (λ = 0.001) to the convolutional layer. The amount by which filters shift, the strides, is set to 1 for the convolutional layer and 2 for the max pooling layer. The input layer was formed by two channels of 1D data that correspond to the AMPA and GABA time series simulated by the point-neuron network. Instead of using data of the whole simulation (3000 ms), we split time series into multiple chunks (i.e., samples) of 100 ms, a window size that we found convenient to improve estimation accuracy of the CNN. Nodes of the output layer predict segments of the EEG signal at each 100-ms window.

The CNN was trained by first-order gradient descent (Adam optimizer [52]) with default parameters as those provided in the original paper. We defined the loss function for training as the mean squared error (MSE) between the predicted and the true values of the EEG. To monitor training, we employed the MSE and also the mean absolute error (MAE) and the coefficient of determination, $R^2$. The CNN is trained for a sufficiently large number of epochs, 100 epochs, to ensure convergence of the error metrics. To train and test the CNN, we use the same datasets generated for optimizing parameters of the current-based proxies, as described above.

## Band-limited inputs across lower frequency bands

In Fig 8, we simulated an increase of low-frequency power in the thalamic input by superimposing a band-limited signal $s(t)$ to the constant baseline term $v_0$. The spike train activating the thalamocortical synapses was thus given by the positive part of a Poisson process with a time-varying rate $v_{ext}(t) = [v_0+s(t)]_+$. To generate the band-limited signal $s(t)$, we created a white Gaussian noise with zero mean and we adjusted the variance of the noise to produce a relative increase of low-frequency power that was within the empirical range observed in real EEG/MEG data. Frequencies in selected low-frequency bands were then extracted by a 2$^{nd}$ order Butterworth bandpass filter. To separate the narrow band oscillations from the aperiodic power-law spectral component, we applied the FOOOF algorithm [53] to the EEG spectra. The aperiodic-adjusted power was computed by subtracting the aperiodic fit from the power spectrum [53,54] as $\log_{10} PSD_{osc}-\log_{10} PSD_{1/f}$, where $PSD_{osc}$ and $PSD_{1/f}$ were the PSD values of the oscillatory peak and the aperiodic power law component respectively calculated at the center frequency of each frequency band.

## Computation of ground-truth EEGs and proxies on the New York head model

To compute the ground-truth EEG with the New York head model [66], we used the current dipole moments generated by multicompartment neurons of two different subnetworks (S4D Fig) from the unconnected network model shown in Fig 9E (for a more detailed description of this approach, see [35]). First, all single-cell current dipole moments from the given subnetwork where summed into a single population current dipole moment. We placed these population current dipole moments in two different locations spatially separated and orthogonally oriented on the folded cortical surface of the New York head (S4B Fig). Topographic maps generated from EEG electrodes were computed for each current dipole moment individually and then summed up together to produce the total EEG map.

We then checked if our EEG proxies could be used to predict current dipole moments. We found that the EEG signal calculated at top of the head in the four-sphere head model, resulting from a current dipole directly below the electrode, was in fact just a scaling of the dominant component (the component aligned with the depth axis of the cortex) of the original current dipole (S3 Fig). This meant that the proxies developed for the EEG signal at the top of the four-sphere head model, were in fact equally valid as proxies for the (normalized) dominant component of the population current dipole moment. Based on this observation, we generated EEG maps from our proxies (in our example, we used *ERWS2*) by plugging their outputs into the dominant component of the current dipole moment and recalculating EEGs on the New York head model.

## Analysis of network states

To characterize the different network states of activity in the point-neuron network at the level of both single neurons and populations, we employed the descriptors developed by Kumar and collaborators for conductance-based point-neuron networks [44].

**Synchrony.**   We quantified the synchrony of the population activity in the network as the average pairwise spike-train correlation from a randomly selected subpopulation of 1000 excitatory neurons. The spike trains were binned in non-overlapping time windows of 2 ms.

**Irregularity.**   Irregularity of individual spike trains was measured by the coefficient of variation (the ratio of the biased standard deviation to the mean) of the corresponding interspike interval (ISI) distribution. Low values indicate regular spiking; a value of 1 reflects Poisson-type behavior. The irregularity index was computed for all excitatory neurons.

**Mean firing rate.**   The mean firing rate was estimated by averaging the firing of all excitatory cells, and was calculated with a bin width of 1 ms.

## Post-processing and spectral analysis

The z-scored EEG signals and proxies were resampled by applying a fourth-order Chebyshev type I low-pass filter with critical frequency $f_c$ = 800 Hz and 0.05 dB ripple in the passband using a forward-backward linear filter operation and then selecting every 10th time sample. The estimate of the normalized power spectral density (normalized PSD) was computed using the Fast Fourier Transform with the Welch's method, dividing the EEG z-scored data into eight overlapping segments with 50% overlap.

## Numerical implementation

Here we summarize the details of the software and hardware used to generate the results presented in this study. Point-neuron network simulations were implemented using NEST

v2.16.0 [89]. EEG signals were computed using LFPy v2.0 [39] for the four-sphere head model and the LFPykit module [90] for the New York head model. Simulations of multicompartment model neurons using NEURON v7.6.5 [91]. The CNN is constructed based on the machine-learning library Keras v2.3. The source-code structure relies on the freely available, object-oriented programming language Python (v3.6.9). Every simulation was parallelized using the following high-performance computing infrastructures: a 60-CPU 256-GB and a 48-CPU 128-GB dedicated servers, as well as the Franklin HPC cluster, all three at the Istituto Italiano di Tecnologia (IIT). We also used the Stallo high-performance computing facilities (NOTUR, No. NN4661K, the Norwegian Metacenter for Computational Science). Simulations of the point-neuron network were performed based on thread parallelism implemented with the OpenMP library. Network simulations with NEURON used distributed computing built on the MPI interface. Computation time for completing simulations of both network models and the post-processing of results was 2 hours on average for each experimental condition.

## Supporting information

**S1 Fig. Performance of proxies for a heterogeneous population of pyramidal cells.** In the same simulation, the "NMC L2/3 PY, clone 9" morphology was randomly assigned to half of the pyramidal-cell population and the "NMC L2/3 PY, clone 0" morphology to the other half. Colors used for proxies are the same used in Fig 4.
(TIF)

**S2 Fig. Learned filters of the convolutional layer and illustration of time shifts applied by the CNN to AMPA and GABA input currents.** Examples of weights learned by four filters of the convolutional layer, depicted both in the time (A) and frequency domains (B) for the AMPA and GABA inputs. (C) Examples of the CNN outputs in response to unit impulses applied either to the AMPA or GABA inputs. (D) Histograms of time shifts applied to the AMPA and GABA inputs for all combinations of impulses. Each time shift is computed as the difference between the time when the impulse is applied and the time in which the absolute response of the CNN reaches its maximum.
(TIF)

**S3 Fig. Comparison between EEG and the dominant component of the current dipole moment at the top of the head.** Example of time sequences of EEG (black line) and the dominant component of the current dipole moment, $P_z$ (red dashed line), at the top of the four-sphere head model, computed both on the multicompartment model network.
(TIF)

**S4 Fig. Computation of EEGs on the New York head model.** (A) Distribution of EEG electrodes and brain surface of the New York head model [66]. The black line represents the cortical cross-section where current dipoles were placed. (B) Current dipoles of the two subnetworks, S0 and S1, (represented by arrows of different colors) positioned in the cortical cross-section. (C) Topographic maps generated from EEG electrodes projected onto two-dimensional simplified plots of the head model. The ground-truth EEG maps are plotted on the left column and EEG estimations of the *ERWS2* proxy on the right column. The closest EEG electrode of each current dipole is plotted as a spot in the same color of the corresponding current dipole. The EEG electrode selected to show time traces of the EEG signal is depicted as a gray spot. (D) Dominant component of the current dipole moment ($P_z$) for the two subnetworks. (E) Time sequences of the ground-truth EEG and the *ERWS2* proxy registered at the electrode shown in panel C. (F) Time traces of ground-truth EEG and the *ERWS2* proxy

registered at different electrodes located in positions indicated by the gray spots.
(TIF)

## Acknowledgments

We are grateful to R. Gao for useful suggestions and M. Libera for technical support.

## Author Contributions

**Conceptualization:** Pablo Martínez-Cañada, Stefano Panzeri.

**Data curation:** Pablo Martínez-Cañada.

**Formal analysis:** Pablo Martínez-Cañada, Torbjørn V. Ness.

**Funding acquisition:** Pablo Martínez-Cañada, Gaute T. Einevoll, Tommaso Fellin, Stefano Panzeri.

**Investigation:** Pablo Martínez-Cañada.

**Methodology:** Pablo Martínez-Cañada, Torbjørn V. Ness, Gaute T. Einevoll, Tommaso Fellin, Stefano Panzeri.

**Resources:** Tommaso Fellin, Stefano Panzeri.

**Software:** Pablo Martínez-Cañada, Torbjørn V. Ness.

**Supervision:** Gaute T. Einevoll, Tommaso Fellin, Stefano Panzeri.

**Visualization:** Pablo Martínez-Cañada, Torbjørn V. Ness.

**Writing – original draft:** Pablo Martínez-Cañada, Stefano Panzeri.

**Writing – review & editing:** Pablo Martínez-Cañada, Torbjørn V. Ness, Gaute T. Einevoll, Tommaso Fellin, Stefano Panzeri.

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
