## [Decision Letter · Decision Letter 0]

4 Dec 2020

Dear Dr Panzeri,

Thank you very much for submitting your manuscript "Computation of the electroencephalogram (EEG) from network models of point neurons" for consideration at PLOS Computational Biology.

As with all papers reviewed by the journal, your manuscript was reviewed by members of the editorial board and by several independent reviewers. In light of the reviews (below this email), we would like to invite the resubmission of a significantly-revised version that takes into account the reviewers' comments.

We cannot make any decision about publication until we have seen the revised manuscript and your response to the reviewers' comments. Your revised manuscript is also likely to be sent to reviewers for further evaluation.

Sincerely,

Daniele Marinazzo

Deputy Editor

PLOS Computational Biology

Reviewer's Responses to Questions

**Comments to the Authors:**

Reviewer #1: In this manuscript, Martinez-Cańada and colleagues employ the “hybrid” framework to simulate EEG more efficiently by decoupling the spiking dynamics generation (using point neuron Brunel E-I networks in different regimes) and the generation of electric fields using morphologically realistic, multi-compartmental neurons. The goal of the work is to find suitable proxies measurable in the spiking neural network (SNN), such as population firing rates or synaptic currents, that can be combined to approximate the “ground-truth” EEG signal generated from the biophysical model. The authors compare an array of such proxies and find that optimized lagged summation of the E and I currents performs the best (for linear proxies) across 1) all network regimes, 2) neuron morphologies and dendritic distribution of synapses, 3) EEG electrode positions, and 4) for non-stationary network states triggered by a Gaussian pulse thalamic input. They also test the performance of nonlinear proxies by training a shallow convolutional neural network on the SNN synaptic currents, and find small but consistent improvements over the linear proxies.

Overall, I find this paper and this line of work to be important, since the ability to make physiological inferences from non-invasive human brain signals, such as the EEG, would open up many avenues of research connecting cognitive neuroscience with computational and systems neuroscience. The simulations and analyses performed in this manuscript seem to be of high quality, very comprehensive, and well-documented, and they generate additional insights on the relationships between the different network measurements, such as the delay between firing rate vs. synaptic currents, as well as their relationship with the EEG. That being said, I’m not sure if this manuscript communicates significant advances in insights and tools over what’s already been presented in Mazzoni et al 2015 and Hagen et al 2016 (both of which I am a huge fan of) from the perspective of EEG modeling, due to the validity of the “ground truth” EEG model and a lack of emphasis (in the writing) on the insights that ARE generated from these computational experiments. The latter can be amended in a straightforward manner, for which I have some suggestions writing-wise. The former is discussed as a major concern below, for which I don’t really have recommendations for as a reviewer (other than to conduct non-trivial additional experiments), but I leave it to the editor to evaluate the suitability of this manuscript for publication considering those gaps.

The key idea of this paper is that one may use spiking neural network outputs to directly simulate the EEG without computationally expensive biophysical simulations, and these experiments check if and when proxies are good substitutes for the full simulation. As such, my main concern is that the value of this work hinges on the validity of the “ground-truth” EEG model: the more accurate the ground truth EEG model is to real EEG signal generation mechanisms, the more valuable it would be to use an SNN proxy directly. On the other hand, if the the ground truth EEG model does not accurately describe known biophysical mechanisms, then it doesn’t matter if one can find perfect proxies using SNN measures to approximate the “ground-truth” simulation, since it would not be useful in realistic situations, especially in use cases suggested in the discussion, i.e., inference. At the very least, this point should be clearly acknowledged in the discussion section as a limitation, beyond what it currently states in terms of extending to more complex head models. Given that, I feel that both components of the hybrid model suffer weaknesses when compared to real EEG, and I’ve listed the major ones below:

- while the models presented in this work makes some advances from Mazzoni et al 2015, such as including a broader set of Brunel network regimes, I find that the spiking dynamics—and as a result the simulated ground-truth EEG—are not representative of signatures often found in EEG. In the EEG, there is typically strong low-frequency oscillations, such as alpha (10Hz) or beta (20Hz). In addition, the power spectrum is almost always 1/f up to very low-frequencies, and often Lorentzian-like (1/f with plateau at low frequencies). While the simulated EEG here can oscillate at 50Hz in the SR state, which can sometimes be observed in the EEG as gamma oscillations, both the time series and the power spectra do not in general look like realistic EEGs, especially in the SI state PSD (Fig. 1I). I believe this is partially due to the limitations of the Brunel network, as it lacks recurrent dynamics between the cortical and thalamic populations, though that is thought to be a crucial generator of prominent EEG features, such as the alpha rhythms (e.g., Halgren et al., PNAS 2020). Setting up a recurrent thalamocortical network may be outside the scope of the current study, but I suggest at least an experiment using oscillatory thalamic input at 10Hz, similar to what the authors have done with the Gaussian packet to simulate an ERP-like signal.

- the current study only considers AMPA and GABA current, while slower currents, such as NMDA and calcium transients, most likely have strong contributions to the scalp EEG as well, especially since it is thought that low-frequency components of cortical activity sum constructively to make up the EEG. Again, this requires modification to the Brunel network in non-trivial ways, but it is not a guarantee that the current results will hold, or simply extrapolate linearly, with more complex currents, and should at least be mentioned in the discussion.

- neither the SNN nor the biophysical column have distance-dependent connections. I’m not sure how big of a deal this is, especially since the point neuron identities are randomly assigned to the biophysical neurons, but spatial correlation of activity could be another factor that would change how the sources are summed to get the EEG.

- the ground-truth EEG signal is assumed to come from just a single cortical column, and while this is closer to the truth for the extracellular LFP, it’s not accurate as a model of the EEG. It’s more likely that the signal an EEG sensor picks up is a complex summation of many such cortical columns across centimeters, oriented in different directions due to cortical folding, and even from heterogeneous cell-types, ratios of E/I cells, and neuronal morphologies (see. e.g., Wang Nat Rev Neurosci 2020). While a full-cortex spiking neural network model with heterogeneous populations is obviously out of the scope, it would be nice to see multiple interacting cortical populations spaced farther apart to mimic source mixing in EEG, especially considering the hybrid model studied in Hagen et al CerCor 2016. Or, at the very least, acknowledge the need to consider such factors in the discussion.

- similarly, the head model is a simple 4-layer spherical model with differential resistance, which is arguably the only addition to the model from Mazzoni et al PLOSCB 2015. The authors state in the discussion that more complex head models can be easily incorporated, since the simulated EEG sensors essentially pick up a scaled version of the dipole in the z-direction. However, as one of the main result is that the proxy perform differently when the sensor is at a different angle with respect to the population, and that cortical folding would imply the summation of sources from differentially oriented pyramidal populations, I’m not convinced that ERWS, for example, would apply straight out the box in that situation. It would be more convincing to show this with several columns, or discussed as a limitation.

- relating to the previous point, I think it’s not clear until much later on (first time in L191) that the EEG model is meant to be a rodent model (with a downscaled and smooth cortex). I think it would orient the reader to have the correct expectation if this was mentioned as early as possible, potentially even in the abstract.

some stand-alone comments:

- I believe that the manuscript would be improved if specific findings are more explicitly highlighted, perhaps even in the abstract. For example, the fact that linear combination of AMPA/GABA currents is better than firing rate or Vm (e.g., L674) is very important for somebody simulating EEG signals from spiking networks, since firing rate is very commonly (and perhaps mistakenly) used as a surrogate. Also, it’s quite interesting that the optimized parameters for ERWS proxies are state-dependent, even though the physical shapes of the neurons themselves are unchanged, which suggests frequency-dependent filtering characteristics of the membrane or synapse.

- computing the R^2 on the time series emphasizes low-frequency fluctuations due to 1/f scaling of EEG signals, does this potentially obfuscate important differences in performance for the different metrics?

- R^2 = 0.9 is used as a benchmark throughout the study, but why, and what is considered “good”?

- is the PSD R^2 computed on log power or raw power? To complement the time series R^2 comparison, perhaps it would be good to compute R^2 on the logged power to de-emphasize the low frequencies.

- worth discussing why Fig1G (R^2 on time series) and J (R^2 on PSD) are so different

- L361: why are FR, sum(I) and Vm so bad? Apologies if I missed the extended discussion on this but I think this is an interesting point, further supporting the insight that the EEG is more about absolute current fluctuations, and less so about net input

- in my opinion, Fig 8 complicates the main message, which I interpret to be that linear proxies are sufficient for estimating the EEG, and thus can be moved to be supplemental as additional analyses.

To summarize, while the EEG ground-truth model realism is a matter of the authors’ decisions, and one can always include more details at the cost of computational complexity, there are several key factors that should be considered, or at least addressed and justified as to why they are omitted. On the other hand, the dynamics produced in the ground truth simulation do not resemble realistic EEG, and while I don’t think a computational study MUST include empirical data, I do think comparison to some real data, even just at a surface level, would ground this work, especially considering the implications suggested in the discussion section regarding inference. I absolutely agree that computational models serve as an important tool for understanding the EEG and allows for physiological inferences. However, while the discussion currently (L730-744) is a nice exposition of the potential uses in theory, since this paper does not actually do it with real data, it’s unclear then from the paper itself how the current framework is suppose to be used for a naive reader (like if an EEG practitioner that wants to use the simulations for inference). So the claims read a little empty. Furthermore, some key references that work on those approaches, such as the Human Neocortical Solver (Neymotin et al eLife 2020), are missing.

Richard Gao, PhD

Department of Cognitive Science, UCSD

Reviewer #2: Building upon previous work to introduce a proxy for the LFP generated by a network of point neurons, in this work the authors adapt this approach to develop a new proxy for the EEG generated by a network of point neurons. As in previous work, the authors feed spikes generated in the point neuron network to a multicompartment, morphological model in order to compare LFP generated by the point neuron network to a ground truth. They use this approach to develop a new proxy suitable for the EEG generated from a network of neurons. This new proxy surpasses previous work, in that this proxy remains valid across a wide range of network states. Finally, they extend their approach using a shallow CNN to understand how much more variance could be explained by more complicated nonlinear proxies.

This manuscript reports well-executed and important work. The utility of a well-motivated EEG proxy in the field of neuroscience is great. There are a few revisions and minor suggestions to address in this round of review.

MAJOR

- line 153: The parameters for the LIF model are well documented in the tables; however, it would be helpful to have a little information in the main text about the approximate scale (i.e. number of neurons and synapses) when introducing the LIF model. This also raises an important point: what network scale is sufficient to faithfully model the EEG? Would the network dynamics change significantly for more than 5,000 neurons, and would this have an impact on the simulated EEG? Lastly, in the case that a larger network would need to be considered, how could the EEG proxy integrate signal across points in space?

- line 448: It is useful to note that the parameter values for the proxies were fixed across different morphologies. This increases confidence that the EEG proxy will generalize well across different morphologies. One point, however, is that the same morphology is applied for the pyramidal cells and the interneurons in the full simulation. What effect could heterogeneity in neuronal morphologies have on the proxy comparison? Is there some way to show that this will be negligible?

- Fig 6: The results from this point are important but somewhat difficult to interpret. What do these distances mean in terms of typical EEG placements in a human experiment?

- Lastly, this proxy has clear potential to shed light on important signals recorded in EEG, for example sleep rhythms. Could this proxy already give some insight into oscillations recorded in the sleep EEG?

MINOR

- In general, the notation in the equations could use improvement. Several style guides caution against multiletter abbreviations for mathematical variables (cf. Physical Review Style and Notation Guide, Section C, Point 4), as it makes equations less clear.

- line 539: Are these "parameters" or "state variables"?

- line 562: What is the unit of the standard deviation?

- Fig 7D: What is the unit of "True values" and "Predictions"?

- line 606: This sentence is unclear. Is there a typo?

**Have all data underlying the figures and results presented in the manuscript been provided?**

Reviewer #1: Yes

Reviewer #2: None

PLOS authors have the option to publish the peer review history of their article (what does this mean?). If published, this will include your full peer review and any attached files.

Reviewer #1: **Yes: **Richard Gao

Reviewer #2: No
---

## [Decision Letter · Decision Letter 1]

16 Mar 2021

Dear Dr Panzeri,

Thank you very much for submitting your manuscript "Computation of the electroencephalogram (EEG) from network models of point neurons" for consideration at PLOS Computational Biology. As with all papers reviewed by the journal, your manuscript was reviewed by members of the editorial board and by several independent reviewers. The reviewers appreciated the attention to an important topic. Based on the reviews, we are likely to accept this manuscript for publication, providing that you modify the manuscript according to the review recommendations.

Sincerely,

Daniele Marinazzo

Deputy Editor

PLOS Computational Biology

Daniele Marinazzo

Deputy Editor

PLOS Computational Biology

[LINK]

Reviewer's Responses to Questions

**Comments to the Authors: **

Reviewer #1: dear all,

many thanks to the authors for engaging so constructively with my comments. I read with great interest the additional experiments and insights they generated, as well as the extensive explanations the authors provided (e.g., on conductance vs. current based models and their implications near reversal potential, timing differences of currents vs. Vm, etc.), which I learned a lot from. As previously agreed upon, these set of experiments are satisfactory to me, and I believe they extend the study in many directions than originally presented, and I hope the authors feel the same after the great efforts after the first round of reviews. The additional text also provide better framing and more explicit acknowledgement of the limitations (and therefore future opportunities) of this work, and I agree that this is an excellent step towards human EEG signals with biophysical modeling.

just one small note: the PSDs in figure 8 look a bit strange, in that none of them show clear oscillatory peaks except in perhaps the delta/theta range, while the aperiodic-adjusted power was apparently non-trivial (as the authors state, compared to results from human data in literature). I'm wondering if there was potentially a mix up with the PSDs plotted? If not, it is curious that the network then somehow filtered out the input frequencies (at least by visual inspection from the PSDs). This does not hinder acceptance in my view, but could be good to discuss, especially as curious readers may wonder where those oscillatory inputs have gone.

again, excellent work and it was a pleasure to be a small part of this process 

best,

richard gao

Reviewer #2: In their revised manuscript, the authors have fully addressed the concerns raised in the first round of review.

**Have all data underlying the figures and results presented in the manuscript been provided?**

Reviewer #1: None

Reviewer #2: Yes

PLOS authors have the option to publish the peer review history of their article (what does this mean?). If published, this will include your full peer review and any attached files.

Reviewer #1: **Yes: **Richard Gao

Reviewer #2: No

Figure Files:

Data Requirements:

Reproducibility:

References:

---

## [Editor Report · Decision Letter 2]

18 Mar 2021

Dear Dr Panzeri,

We are pleased to inform you that your manuscript 'Computation of the electroencephalogram (EEG) from network models of point neurons' has been provisionally accepted for publication in PLOS Computational Biology.

Best regards,

Daniele Marinazzo

Deputy Editor

PLOS Computational Biology

Daniele Marinazzo

Deputy Editor

PLOS Computational Biology

---

## [Editor Report · Acceptance letter]

29 Mar 2021

PCOMPBIOL-D-20-01969R2 

Computation of the electroencephalogram (EEG) from network models of point neurons

Dear Dr Panzeri,

I am pleased to inform you that your manuscript has been formally accepted for publication in PLOS Computational Biology. Your manuscript is now with our production department and you will be notified of the publication date in due course.

With kind regards,

Katalin Szabo
